# Structure of the active form of human origin recognition complex and its ATPase motor module

Ante Tocilj[1,2,3], Kin Fan On[1,2,3], Zuanning Yuan[4,5], Jingchuan Sun[4†], Elad Elkayam[1,2,3], Huilin Li[4,5†], Bruce Stillman[3*], Leemor Joshua-Tor[1,2,3*]

[1]W. M. Keck Structural Biology Laboratory, Cold Spring Harbor, New York, United States; [2]Howard Hughes Medical Institute, Cold Spring Harbor, New York, United States; [3]Cold Spring Harbor Laboratory, Cold Spring Harbor, New York, United States; [4]Biology Department, Brookhaven National Laboratory, New York, United States; [5]Department of Biochemistry and Cell Biology, Stony Brook University, Stony Brook, United States

**Abstract** Binding of the Origin Recognition Complex (ORC) to origins of replication marks the first step in the initiation of replication of the genome in all eukaryotic cells. Here, we report the structure of the active form of human ORC determined by X-ray crystallography and cryo-electron microscopy. The complex is composed of an ORC1/4/5 motor module lobe in an organization reminiscent of the DNA polymerase clamp loader complexes. A second lobe contains the ORC2/3 subunits. The complex is organized as a double-layered shallow corkscrew, with the AAA+ and AAA+-like domains forming one layer, and the winged-helix domains (WHDs) forming a top layer. CDC6 fits easily between ORC1 and ORC2, completing the ring and the DNA-binding channel, forming an additional ATP hydrolysis site. Analysis of the ATPase activity of the complex provides a basis for understanding ORC activity as well as molecular defects observed in Meier-Gorlin Syndrome mutations.

**\*For correspondence:** stillman@cshl.edu (BS); leemor@cshl.edu (LJ)

**Present address:** †Van Andel Research Institute, Grand Rapids, United States

**Competing interests:** The authors declare that no competing interests exist.

## Introduction

The first step in genome replication, the binding of the Origin Recognition Complex (ORC) at origins of DNA replication, triggers a series of highly coordinated steps leading to the assembly of pre-replicative complexes (pre-RCs) in a process that involves CDC6 binding to ORC (*Bell and Labib, 2016*). ORC and CDC6 then function as an ATP-dependent assembler that first recruits a ring-shaped MCM2-7 hexamer with bound Cdt1 to DNA, and then loads a second MCM2-7 hexamer in a head-to-head orientation, whereby this double hexamer is topologically linked to double-stranded DNA (reviewed in [*O'Donnell et al., 2013*; *Siddiqui et al., 2013*; *Bell and Labib, 2016*]). These MCM2-7 double hexamers mark the location of potential origins of DNA replication that can be activated during S phase to replicate the genome.

In *Saccharomyces cerevisiae*, the pre-RC includes the two head-to-head associated Mcm2-7 hexamers with each hexamer destined to become a part of the Cdc45-Mcm2-7-GINS (CMG) helicase at each replication fork during S phase (*Abid Ali and Costa, 2016*). Regulated assembly and disassembly of the pre-RC ensures that the genome replicates only once during each cell division cycle (*O'Donnell et al., 2013*; *Siddiqui et al., 2013*). In budding yeasts, ORC is a stable complex consisting of five AAA+ (ATPases Associated with diverse cellular Activities) related subunits and another subunit, Orc6. Orc1, Orc4 and Orc5 are bona-fide members of a large class of AAA+ proteins that contain the characteristic Walker-A and Walker-B motifs, but the only active ATPase in *S. cerevisiae*

ORC is the Orc1 subunit with its adjacent Orc4 subunit that contributes an activating arginine finger and other residues required for ORC function in vivo (*Loo et al., 1995*; *Klemm et al., 1997*; *Bowers et al., 2004*). Mutations in the Walker-A motif of *S. cerevisiae* Orc4 and Orc5 are not essential for cell proliferation (*Loo et al., 1995*). Moreover, Orc2 and Orc3 were predicted to contain an AAA+ fold, but these subunits do not have the residues that are important for ATP hydrolysis and thus are predicted to be 'AAA+ like' (*Bell et al., 1993*; *Foss and Rine, 1993*; *Micklem et al., 1993*).

The yeast ORC binds to origin DNA throughout the cell division cycle, even during mitotic exit. During G1 phase, it recruits the Orc1-related initiator protein Cdc6, another AAA+ ATPase, to the origin DNA, creating an extended footprint over the entire origin (*Diffley et al., 1994*; *Liang et al., 1995*; *Speck et al., 2005*). In contrast, in human cells, ORC binding to chromosomes is dynamic, with the first step being the binding of ORC1 to mitotic chromosomes and the subsequent assembly of ORC onto chromatin during the next G1 phase (*Okuno et al., 2001*; *Siddiqui and Stillman, 2007*; *Wu et al., 2014*; *Kara et al., 2015*; *Abid Ali and Costa, 2016*). During entry into S phase, the human ORC1 subunit is degraded (reviewed in [*DePamphilis 2003*]).

A detailed molecular understanding of the ORC-CDC6 ATPase loading machine and its motor function is essential to decipher how it works to initiate replication. Several low-resolution structures of ORC from *Drosophila melanogaster* (*Clarey et al., 2006*; *Bleichert et al., 2013*) and *Saccharomyces cerevisiae* (*Speck et al., 2005*; *Sun et al., 2012*) are available, but they show different architectures. Recently, a high-resolution crystal structure of the *Drosophila melanogaster* ORC (DmORC) was reported (*Bleichert et al., 2015*). This structure contains truncated versions of ORC1, ORC2, and ORC3, full-length ORC4 and ORC5, and a small fragment of ORC6. The overall shape of the crystal structure matches the shape observed in the low-resolution EM structures of this complex. The DmORC was captured in a state that is inactive for ATP hydrolysis by ORC1, the most critical ATPase subunit. This is because the interface between ORC1 and ORC4 and thus the ATP-binding site is disrupted, even though one of the crystal forms reported bound nucleotide. This state of the complex would have to undergo a substantial conformational transition in order to become functionally active for ATP hydrolysis, as noted by the authors.

Here, we report the structure of human ORC in a functionally active, ATP-hydrolysis ready state, providing insight into ATP-dependent protein loading as well as DNA and CDC6 binding. We also examine the effect of heretofore uncharacterized Meier-Gorlin Syndrome mutations on its molecular function. Meier-Gorlin Syndrome is a primordial dwarfism, microcephalic disorder caused by mutations in genes encoding pre-replicative complex proteins, including ORC1, ORC4, ORC6 and CDC6 (*Bicknell et al., 2011a*, *2011b*; *Guernsey et al., 2011*; *de Munnik et al., 2012b*).

To facilitate structural studies of HsORC, we separated the particle into two lobes, one consisting of an N-terminally truncated ORC1, full-length ORC4 and a C-terminally truncated ORC5 (see Materials and methods, *Figure 1—figure supplement 1*). The other lobe consists of N-terminally truncated ORC2 and full-length ORC3. Both subcomplexes yielded crystals, with the ORC1/4/5 motor module diffracting well to 3.4 Å, and the ORC2/3 subcomplex to 6 Å (*Tables 1* and *2*). Since ORC1/4/5 is comprised of the active ATPase subunits, the inclusion of ATP was critical to maintain its integrity during purification and crystallization. We hereafter refer to the ORC1/4/5 subcomplex as the HsORC motor module. In contrast, HsORC2/3 is composed of AAA+-like subunits, but with no predicted ATPase activity. The two subcomplexes were modeled into a 20 Å cryo-electron microscopy (cryoEM) 3D-reconstruction map to obtain a structural model for the HsORC1-5 complex.

The structure of HsORC reveals a remarkable similarity between two very different ATPases: the replication initiator ORC-CDC6 ATPase and the replication fork DNA polymerase clamp loader (*Kelch et al., 2011*). Both ATPases function at different times during genome replication but load ring-shaped proteins onto double-stranded DNA so that the ring-shaped proteins become topologically linked to the DNA double helix.

## Results and discussion

### The HsORC motor module

The structure of the HsORC motor module was solved by molecular replacement using the DmORC4, DmORC5-AAA+ domain and DmORC1-WHD of the DmORC complex (*Bleichert et al.,*

**Table 1.** Data collection and refinement statistics for HsORC motor module*.

| Data collection | |
|---|---|
| Wavelength (Å) | 0.9793 |
| Resolution range (Å) | 19.88–3.39 (3.52–3.39) |
| Space group | P2$_1$ |
| Unit cell (Å,°) | 120.89 81.14 151.95 90 97.25 90 |
| Total reflections | 148448 (14543) |
| Unique reflections | 39700 (3849 ) |
| Multiplicity | 3.7 (3.8) |
| Completeness (%) | 0.97 (0.96) |
| Mean I/s(I) | 8.75 (1.19) |
| Wilson B-factor (Å2) | 104.85 |
| R-merge | 0.1385 (1.406) |
| R-measure | 0.1619 (1.641) |
| CC1/2 | 0.997 (0.524) |
| CC* | 0.999 (0.829) |
| **Refinement** | |
| Resolution range (Å) | 19.88–3.39 (3.52–3.39) |
| Reflections used in refinement | 39700 (3849) |
| Reflections used for R-free | 1963 (198) |
| R-work | 0.2421 (0.3745) |
| R-free | 0.2811 (0.4015) |
| CC(work) | 0.959 (0.662) |
| CC(free) | 0.923 (0.591) |
| Number of non-hydrogen atoms | 15870 |
| Macromolecules | 15678 |
| Ligands | 192 |
| Protein residues | 1947 |
| RMSD bonds (Å) | 0.004 |
| RMSD angles (°) | 0.72 |
| Ramachandran favored (%) | 95 |
| Ramachandran allowed (%) | 4.6 |
| Ramachandran outliers (%) | 0 |
| Rotamer outliers (%) | 0.57 |
| Clashscore | 3.59 |
| Average B-factor (Å2) | 127.05 |
| Macromolecules | 127.23 |
| Ligands | 112.92 |
| Number of TLS groups | 31 |

*Values in parentheses are for the highest resolution shell.

*2015*) as a search model. The overall architecture of the HsORC motor module resembles a cashew nut (*Figure 1a*). Each ORC subunit is comprised of three domains – the RecA-fold, the α-helical lid and the α-helical winged-helix domain (WHD), although the WHD domain was truncated in ORC5. The RecA-fold domain and the lid together constitute the well-known AAA+ domain. The three RecA domains form a semicircle with ATP nucleotides wedged between them in a classic AAA+

**Table 2.** Data collection and refinement statistics for HsORC2/3*.

| Data collection | |
|---|---|
| Wavelength (Å) | 0.9793 |
| Resolution range (Å) | 20.07–6.00 (6.21–6.00) |
| Space group | P2₁ |
| Unit cell (Å,°) | 87.26 114.96 316.46 90 90.72 90 |
| Total reflections | 52200 (8302) |
| Unique reflections | 15430 (2286) |
| Multiplicity | 3.4 (3.6) |
| Completeness (%) | 95.0 (99.1) |
| Mean I/s(I) | 5.7 (1.0) |
| Wilson B-factor (Å2) | 316 |
| R-merge | 0.183 (>1) |
| R-measure | 0.24 (>1) |
| CC1/2 | 0.991 (0.422) |
| **Refinement** | |
| Resolution range (Å) | 20.07–6.00 (6.45–6.00) |
| Reflections used in refinement | 15179 (1498) |
| Reflections used for R-free | 753 (1498) |
| R-work | 0.3180 (0.3804) |
| R-free | 0.3685 (0.4019) |
| Number of non-hydrogen atoms | 24148 |
| Protein residues | 2944 |
| RMSD bonds (Å) | 0.013 |
| RMSD angles (°) | 1.30 |
| Ramachandran favored (%) | 87 |
| Ramachandran allowed (%) | 12 |
| Ramachandran outliers (%) | 0.8 |

*Values in parentheses are for the highest resolution shell.

oligomerization arrangement (see below). The RecA and lid domains of ORC1 contact the RecA domain of ORC4 as if cupping that domain between the two. The same is true for the RecA and lid domains of ORC4 cupping the RecA domain of ORC5. Remarkably, the three ORC subunits have the exact same organization of the three domains with respect to each other and can be readily super-imposed (*Figure 1b*). The domains arrange themselves into a right-handed helical architecture with a ~52° rotation and 2 Å rise (58° rotation with a 2.05 Å rise from the AAA+ of ORC1 to ORC4; and 46° rotation with a 2.3 Å rise from the AAA+ of ORC4 to ORC5).

The ATPase motor module of HsORC is very reminiscent of the DNA polymerase clamp loader complexes such as Replication Factor C (RFC) in eukaryotes, the bacterial γ-complex, and the T4 bacteriophage Gene44 clamp loader (*Figure 2*). Both the architecture of the subunits and the orga-nization of the domains with respect to each other are remarkably similar, although HsORC is some-what shallower. The rotations between subunits for several clamp loaders range between 56° and 74° with rises in the range of 2.8 to 8.0 Å, excluding a couple of outliers that appear to be in an inac-tive conformation. The clamp loaders are ATPases that perform a single ATPase cycle to pry open a closed DNA polymerase clamp ring in order to load it onto the double helix of DNA (*Kelch et al., 2012*; *Hedglin et al., 2013*), in contrast to the continuously hydrolyzing AAA+ helicases (*Enemark and Joshua-Tor, 2008*). In the case of HsORC, there is also a single ATPase event, either for loading of the double hexamer of MCM2-7 or for resetting the system (*Frigola et al., 2013*).

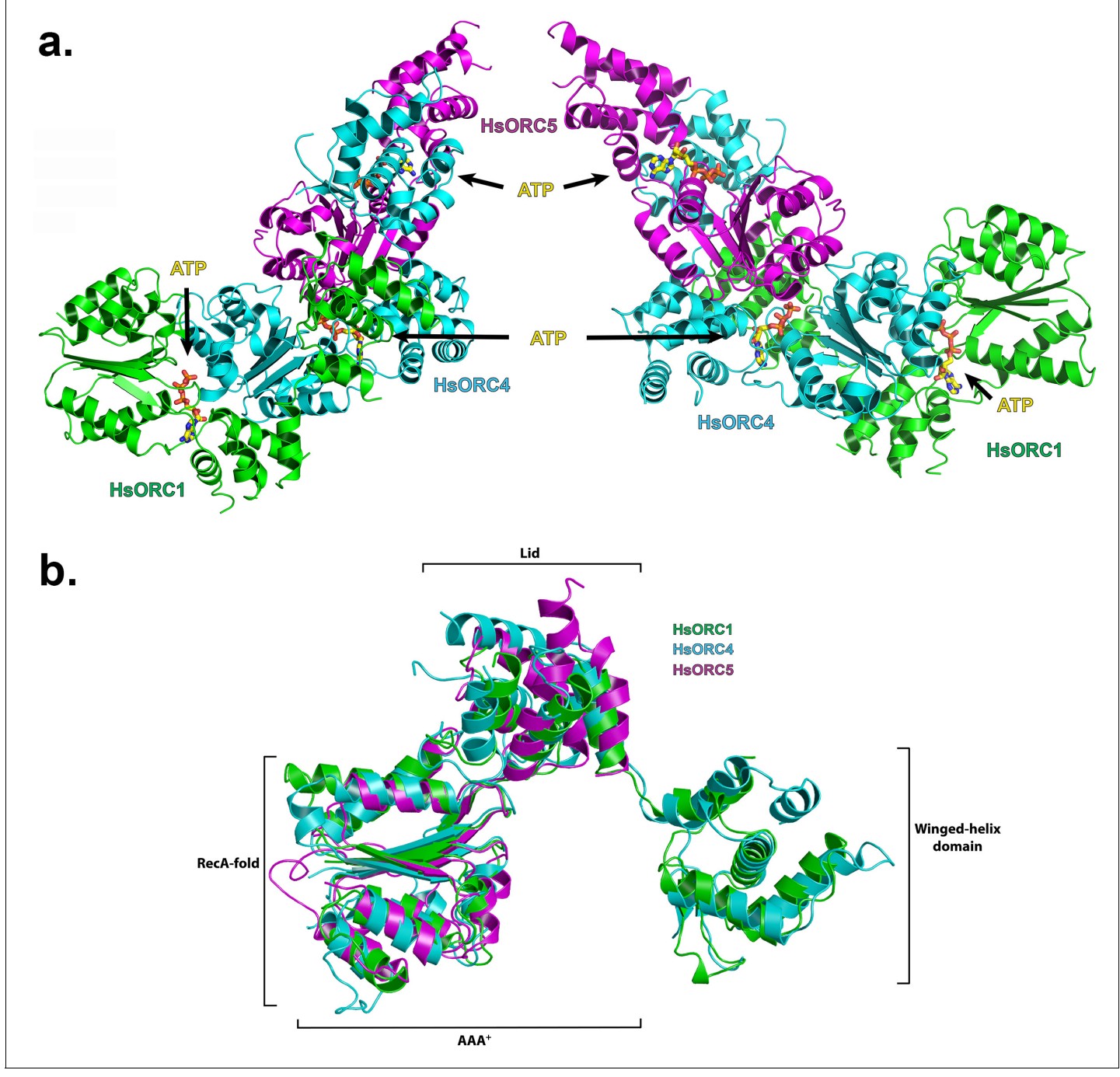

**Figure 1.** Structure of the motor module of HsORC. (**a**) Ribbon diagram of the motor module resembling a cashew nut, with ORC1 in green, ORC4 in cyan and ORC5 in purple. The three ATP molecules nestled between the domains are shown in stick representation. The RecA-fold domains or ORC4 and ORC5 are cupped between the RecA-fold and lid domains of ORC1 and ORC4, respectively. The WHDs of ORC1 and ORC4 are positioned above the RecA-fold domains of ORC4 and ORC5, respectively. (**b**) Superposition of ORC1, ORC4 and ORC5, showing an identical organization between the domains of each ORC subunit.

The following figure supplements are available for figure 1:

**Figure supplement 1.** HsORC constructs used in this study.

**Figure supplement 2.** ATP nucleotides at the motor module subunit interfaces.

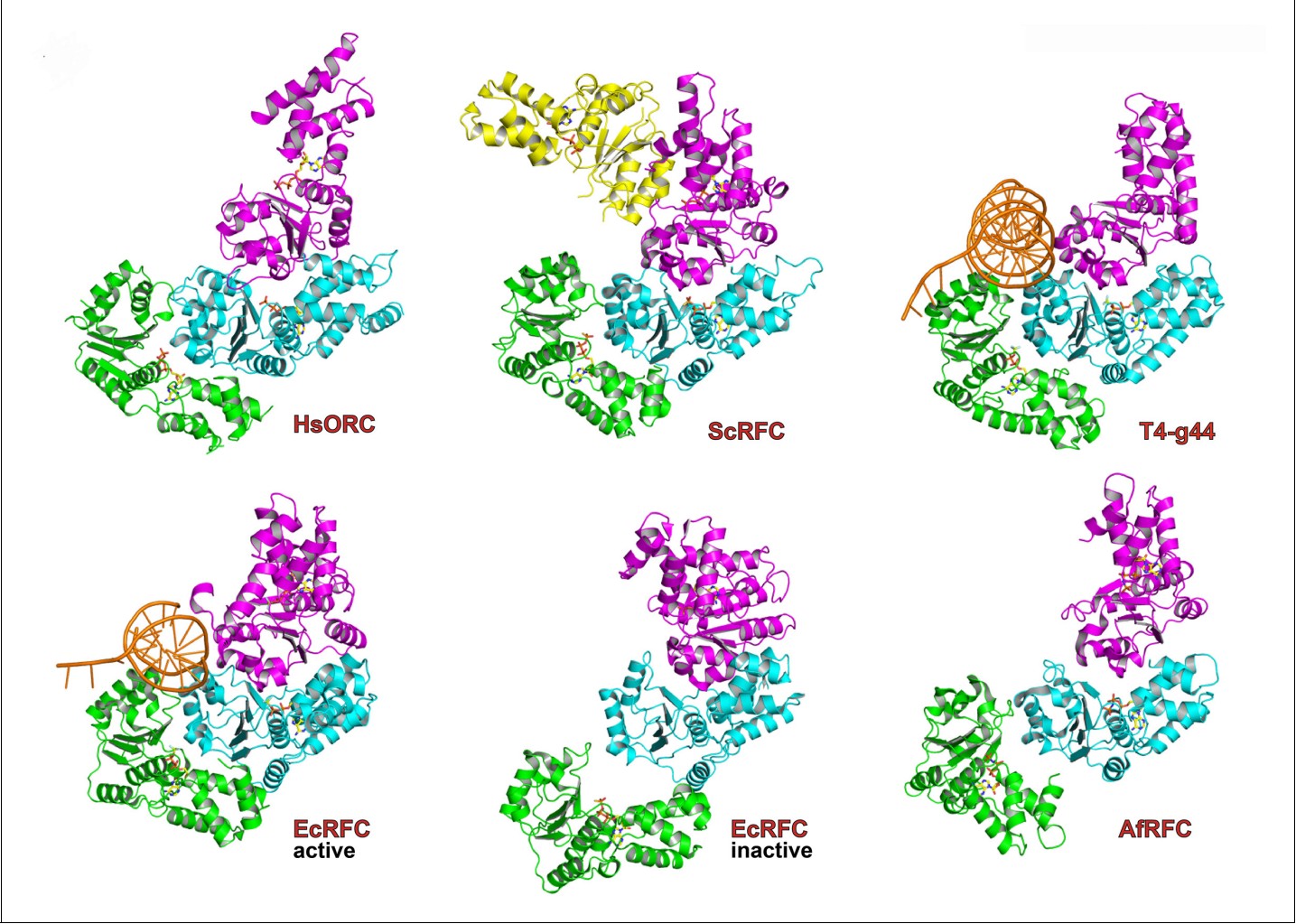

**Figure 2.** Similarity of the HsORC motor module with DNA polymerase clamp loader complexes. Top row: HsORC1/4/5 (this work); yeast RFC-B, -C, -D, -E subunits with nucleotide from the RFC clamp loader bound to the PCNA clamp (*Bowman et al., 2004*) (1SXJ); gp44 subunits with nucleotide from the T4 bacteriophage clamp loader bound to an open clamp and DNA (*Kelch et al., 2011*) (1U60). Bottom row: γ subunits with nucleotide from the *E. coli* clamp loader bound to DNA (*Simonetta et al., 2009*) (3GLF); γ subunits from the inactive *E. coli* clamp loader (*Kazmirski et al., 2004*) (1XXH); small subunits bound to ADPNP from the *Archaeoglobus fulgidus* RFC complex (*Seybert et al., 2006*) (2CHQ). All proteins were aligned based on the subunit depicted in cyan (ORC4 in the case of HsORC).

The HsORC motor module crystallized with ATP, and the nucleotides can be clearly seen in difference electron-density maps between the subunit interface of ORC1/ORC4 and between the interface ORC4/ORC5, as well as on the surface of ORC5 (*Figure 1—figure supplement 2*). Despite the ATPase function of the HsORC motor module, the electron density indicates that ATP hydrolysis had not occurred. We note that the crystals appeared in less than a day and reproducibly disappeared after an additional day if not promptly harvested and frozen. The conformation and placement of the three nucleotides is also quite similar, apart from a rotation of the adenine base with respect to the sugar (the angle of the glycosidic bond) and the absence of an additional subunit that would complete the binding site for ATP in ORC5 (see below).

## The nucleotide-binding sites

As mentioned earlier, ATP hydrolysis by ORC is required for replication in vivo. The nucleotide-binding sites are characteristic of AAA+ ATPases of this family (*Enemark and Joshua-Tor, 2008*; *Hauk and Berger, 2016*). For the nucleotides at the subunit interfaces: Walker-A (P-loop) and

Walker-B motifs from the RecA domain of one subunit line one side of the ATP nucleotide and three basic residues line the other side (*Figure 3a,b*). The first basic residue, the 'trigger', is the sensor-2 residue. Specifically, it is ORC1-R720 for the first (ORC1-ORC4) interface and ORC4-R277 for the second (ORC4-ORC5) interface. The trigger emanates from the lid domain of the subunit that contributes the Walker-A and -B residues to the binding site and is most similar to the F1-ATPase arginine finger (*Kagawa et al., 2004*). The trigger is what differentiates an ATP-bound and ATP-hydrolyzing configuration. The precise positioning of this residue appears to regulate the reactivity of the site, with the ATP-hydrolyzing configuration stabilizing the hydrolysis transition state (*Kagawa et al., 2004*). In ATPases that function continuously, such as helicases, the trigger usually resides on the neighboring subunit to the one harboring the Walker-A and -B motifs, whereas in ATPases such as ORC, and the clamp loader RFC, where there is probably a single hydrolysis event, the trigger resides on the same subunit (*Enemark and Joshua-Tor, 2008*). Factors that interact with the lid domain may have a profound effect on the positioning of the lid domain and therefore the trigger and consequently on the activation of ORC ATPase activity. Interestingly, the ORC1 trigger residue is mutated in a Meier-Gorlin Syndrome (MGS) patient (ORC1-R720Q) who also carries a mutation (ORC1-R105Q) in the amino-terminus that disrupts ORC1's Cyclin E-CDK2 inhibitory activity and histone H4K20me2 binding (*Hossain and Stillman, 2012*; *Kuo et al., 2012*). MGS patients with ORC1 mutations display the most severe phenotype of short stature and microcephaly (*de Munnik et al., 2012a*).

The second basic residue opposite the P-loop is commonly referred to as the R-finger in the AAA+ family and functions as the 'piston', since it moves in and out of the site based on the presence of the γ-phosphate of the nucleotide (*Enemark and Joshua-Tor, 2008*). It is ORC4-R209 at the first interface and ORC5-R143 at the second interface. The third basic residue, the 'tether', forms contacts between the R-finger side of the interface to the Walker-A, -B side. ORC4-R205 performs this function at the first interface as it contacts the Walker-B residue E621 as well as D623 of ORC1. However, a second tether is formed by ORC4-Y174, which interacts with ORC1-E621 as well. ORC4-Y174 is also mutated in three MGS patients (ORC4-Y174C) and is conserved in all ORC4 subunits from yeast to mammals. We suggest that in addition to the well-known R-finger, this tyrosine plays a critical role in sensing the nucleotide state at this critical interface. Indeed, a tyrosine exists at a comparable position in ORC1 (ORC1-Y634) that is highly conserved, but a proline occupies that position in ORC5 (ORC5-P139). We predict that ORC1-Y634 interacts with the CDC6 Walker-B motif (CDC6-E285) in a similar fashion (see below). At the second P-loop interface between ORC4 and ORC5, we identify ORC5-R104 as the tether as it forms a salt bridge with ORC4-E113.

One distinct feature of the Walker-A motif (P-loop) in ORC4 is the presence of an arginine, R69, in the center of the loop (*Figure 3b*). It is completely conserved in mammals, avian, fish and anthropods. It forms a salt bridge with ORC4-E160 of the Walker-B motif. It is also present in the tau subunit of the *E. coli* clamp loader. In tau, this arginine contacts a main-chain carbonyl on a loop of the lid domain (*Simonetta et al., 2009*).

The adenine base is tucked between the RecA domain and the lid of the same subunit, with three hydrophobic residues from the lid forming a greasy side for the base to dock on. Specific H-bonds with the adenine base occur with ORC1 between a tyrosine (Y681) to N6, and a glutamate (E502) to N3. In ORC4, N6 is contacted by a main-chain carbonyl (F38). A main-chain carbonyl is also contacting N6 in ORC5, in addition to H-bonds from R226 to N3 and the hydroxyl of Y182 to N7 (*Figure 3—figure supplement 1*).

The third nucleotide binds to Walker-A and -B motifs on ORC5 (*Figure 3c*), although the Walker-B motif is somewhat different (DKAE, rather than DELD (ORC1) or DEFD (ORC4)). The residue that occupies the same position as the 'trigger' emanating from the lid domain is a lysine – K223. The missing elements needed for ATP hydrolysis, in particular the arginine finger and the tether, would normally be contributed by a neighboring subunit. In the context of the full complex, the RecA fold of ORC3 would occupy the neighboring subunit position (see below), and would be the subunit that would contribute the two other basic residues - the arginine finger (piston) and the tether. The position of the arginine finger on the RecA fold in HsORC1, HsORC4 and HsORC5 is invariable. However, the equivalent residue on the RecA fold of ORC3 is not an arginine but a leucine (HsORC3-L270). There are arginines on either side, however (HsORC3-R98 and HsORC3-R261), that could serve in this capacity for the nucleotide at the HsORC5-HsORC3 interface, if they are positioned properly in the full complex (see below).

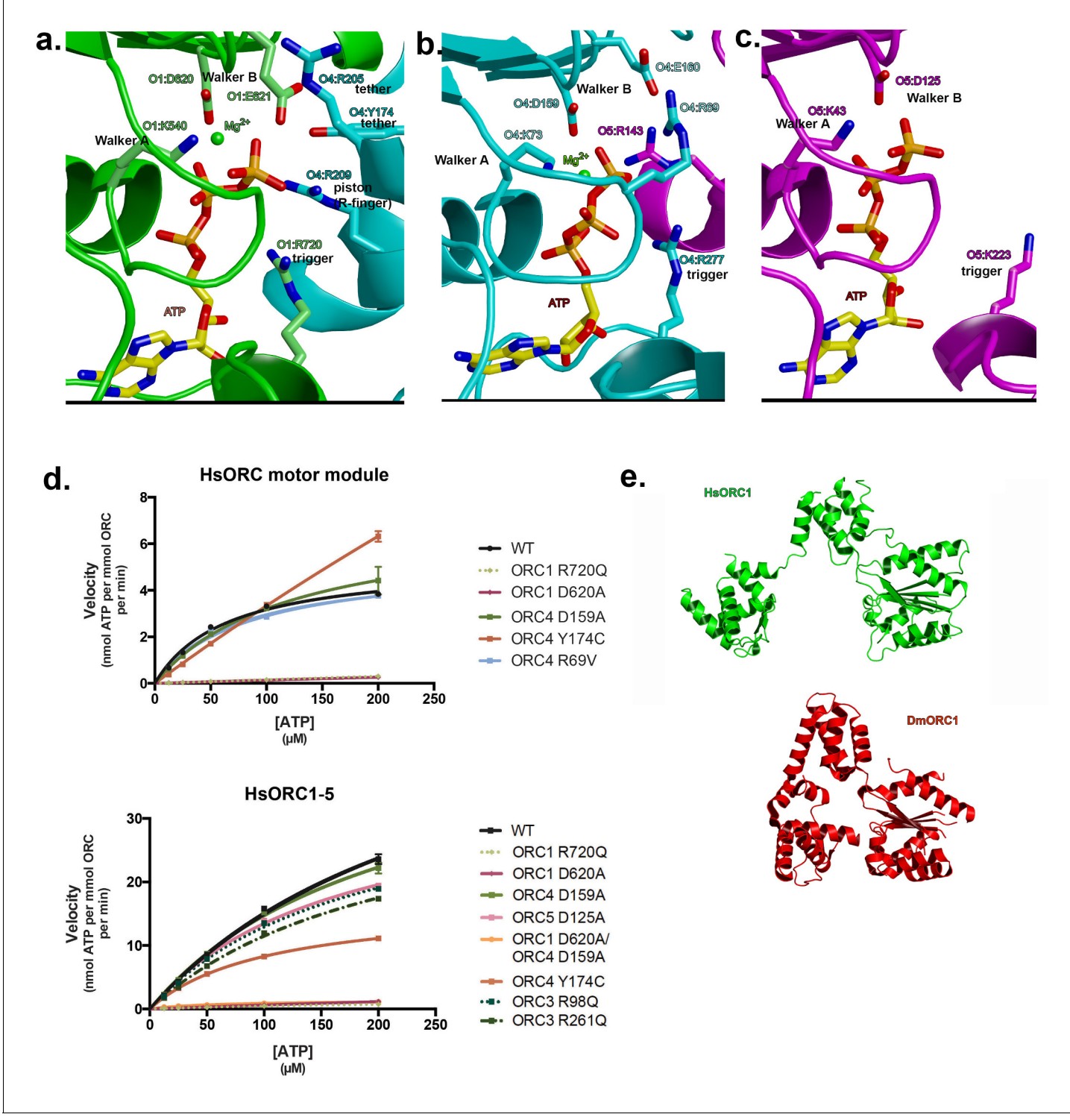

**Figure 3.** Motor module ATPase. (**a**) ATP-binding site at the ORC1/4 interface. (**b**) ATP-binding site at the ORC4/5 interface. (**c**) ATP-binding site of ORC5. ORC1 is shown in green, ORC4 in cyan and ORC5 in purple. ATP and key residues discussed in the text are shown in stick representation. (**d**) Michaelis-Menten curves for ATPase activity of the motor module (top), HsORC1-5 (bottom), for the wild type and several mutants from experiments performed in triplicate. (**e**) Different domain organization of HsORC1 (green) compared with the more compact DmORC1 (red).

The following figure supplements are available for figure 3:

**Figure supplement 1.** Interactions with the adenine base at the ATP-binding sites.

*Figure 3 continued on next page*

*Figure 3 continued*

**Figure supplement 2.** HsORC ATPase activities.

## ORC ATPase levels are altered in Meier-Gorlin syndrome mutants

To test whether the isolated HsORC motor module can hydrolyze ATP, we removed residual ATP by size exclusion chromatography (SEC) and performed ATPase assays on eluted fractions with radiolabeled ATP. The motor module displayed robust ATPase activity, which was independent of DNA binding (*Figure 3d*). To test which of the ATPase sites were functional in the motor module, we mutated several key residues (*Figure 3d*, *Figure 3—figure supplement 2*). Disrupting the ORC1 Walker-B motif (ORC1 D620A) effectively abolished ATPase activity, while mutation of the ORC4 Walker-B motif (ORC4 D159A) had little effect. A double mutant of the Walker-B motif of both ORC1 and ORC4 also abolished activity. We therefore conclude that in the context of the motor module, only the ORC1/4 interface is a functional ATPase. This is in agreement with previous studies in budding yeast where ORC has a single ATPase consisting of ScOrc1 activated by an ScOrc4 R-finger (*Bowers et al., 2004*). The unusual R69 in the Walker-A motif of ORC4 was mutated to test whether its interaction with the Walker-B motif is functional. However, a mutation to valine resulted in similar activity to wild type. Interestingly, the two MGS mutations discussed earlier resulted in very different behaviors with respect to modulation of ATPase activity. ORC1-R720Q abolished ATPase activity of the motor module, and this mutation exists in a single heterozygous individual with a wild-type allele, whereas the ORC4-Y174C mutation in the ORC4 tether, which disrupts its hydrogen bond to an ORC1 Walker-B side chain (ORC1-E621), renders the motor module hyperactive for ATPase activity. Together, these data suggest that properly regulated ATPase levels of HsORC are critical for tissue homeostasis in vivo. The more complete wild-type HsORC particle (containing ORC1-5) has a more robust ATPase activity, as do the ORC4 and ORC5 Walker-B mutated versions. However, the ORC1 Walker-B mutant, the ORC1 Walker-B / ORC4 Walker-B double mutant, as well as the ORC1 MGS mutation R720Q all show very little activity, similar to the situation in the motor module alone (*Figure 3d*, *Figure 3—figure supplement 2*). The ORC4 MGS mutant Y174C has reduced activity (at about 50% of wild type) in the context of ORC1-5. Thus, the hyperactivity of this mutant observed in the context of the motor module alone suggests that binding of ORC2-ORC3 modulates the ATPase activity of the ORC motor. It should be noted that the ORC4-Y174C mutation exists either as a homozygous mutation in one individual or a heterozygous combined with a null mutation in two individuals (*de Munnik et al., 2012b*).

## HsORC1 exists in an active conformation

The architecture of ORC4 and ORC5 is very similar to the architecture of their homologs in the structure of the *Drosophila melanogaster* ORC (DmORC) (*Bleichert et al., 2015*) with a root-mean-square deviation (rmsd) over 389 Cα's of 1.15 Å between the two. However, the architecture and placement of ORC1 is very different. In DmORC, this subunit adopts an unusual conformation, as noted by the authors, whereby the RecA-fold and lid domains, comprising the entire AAA+ domain, are swung out to a different side of the complex, packing against DmORC2, so that it disrupts the ATP-binding site and would therefore be inactive (*Figure 3e*). The functional significance of the DmORC structure is not known. In contrast, the human ORC is active as an ATPase and the RecA-fold and lid domains of HsORC1 form a classic ATPase site as described above, rotated almost 80° from the position in DmORC. The WHDs of HsORC1 and DmORC1, on the other hand, are positioned in an almost identical manner.

## HsORC1-5 forms a double-layered shallow corkscrew

To examine the functional significance of HsORC1 movement, we employed both negative stain and cryo-electron microscopy (cryoEM) to obtain the structure of a more complete HsORC particle. The purified HsORC1-5 described here is a fairly unstable complex, consistent with the dynamic assembly and disassembly of HsORC in vivo. It was purified with ATP and is active for ATP hydrolysis. As a consequence, only a small percentage of the particles viewed by cryoEM were similar to the low-

resolution EM structures of *S. cerevisiae* and the HsORC1-5 obtained from negative stain EM (see Materials and methods). Thus, we reasoned that using this subset of particles in order to obtain a low-resolution structure into which we could fit the crystal structures of the subcomplexes (ORC1/4/ 5 and ORC2/3) was warranted. Cryo-EM data were collected on a Gatan K2 summit direct electron detector in a Titan Krios electron microscope operating at 300 kV, resulting in a ~20 Å resolution 3D density map (*Figure 4a*) (*Figure 4—figure supplement 2–7*). We docked the crystal structure of the HsORC motor module into the EM map, which had an excellent fit. Residual density in the EM map near HsORC1 could be explained by the presence of the 14 kDa Sumo tag on the N-terminus of the truncated HsORC1 construct used for the EM experiment. Although our crystals of the HsORC2/3 complex diffracted only to 6.0 Å, with four complexes in the asymmetric unit, we were able to obtain phases by molecular replacement using the DmORC subunits (*Bleichert et al., 2015*) (*Table 2*). The resulting electron-density map was clear enough to show a significant movement of helices in ORC3 compared to the DmORC structure (*Figure 4—figure supplement 1*). In addition, the position of the WHD of ORC2 would result in significant clashes for two of the four ORC2 copies present in the crystal. This indicated to us that the structures were suitable for docking into the HsORC cryoEM reconstruction. A separate structure of the WHD of HsORC2 is available (PDB code: 5C8H) and a model for the WHD of ORC5 was built based on DmORC followed by energy minimization. These models were docked into the HsORC cryoEM map and subjected to refinement by keeping the motor module ORC1/4/5, the RecA fold of ORC2 and the complete ORC3, the WHD of ORC5 and the WHD of ORC2 as four discrete rigid bodies (*Figure 4b*) (*Video 1*).

HsORC has the appearance of a right-handed corkscrew composed of the AAA+ and WHD of all the subunits, and a 'decorative' handle made of the large helical domain of ORC3 (*Figure 4b,c*). We did not include the small, non-AAA+ ORC6 subunit in this analysis because HsORC1-5 assembly is independent of ORC6 (*Vashee et al., 2001*; *Siddiqui and Stillman, 2007*). The RecA and RecA-like folds of the subunits (ORC 1, 4, 5, 3 and 2) form one layer of the corkscrew, with a little gap between the RecA-like domains of ORC3 and ORC2. The WHDs form a second layer with each domain sitting atop the RecA domain of the neighboring subunit. The structure of HsORC has an open ring architecture with a ~35 Å central channel, in contrast to the closed and more restricted central channel of ~15 Å observed in DmORC (*Bleichert et al., 2015*). This larger opening can now easily accommodate a DNA double helix. Apart from the similarity between ORC4 and ORC5 discussed above, the WHD of HsORC5 is also positioned similarly as in DmORC, and HsORC3 is similar in position and arrangement as in the fly counterpart as is the RecA-like fold of HsORC2. However, in addition to the drastic reorientation of HsORC1 in comparison with DmORC (*Figure 5a*), the position of the WHD of HsORC2 is also clearly very different between the two complexes. In HsORC, it is moved out and up from the central groove and is not interacting with HsORC1, as it would in DmORC if the AAA+ domain of DmORC1 was moved to a similar position as in HsORC (*Video 2*). The consequence of this repositioning is that it opens up the central groove in HsORC considerably (see below) (*Figure 5b*) (*Video 2*). The RecA-like fold of ORC3 is now positioned near the AAA+ domain of ORC5, which is bound by a nucleotide. ORC3 does not have the characteristic side chains positioned at the ATP site to promote hydrolysis. However, even though there is no arginine positioned on the RecA fold in an equivalent position as it is in HsORC1, HsORC4 or HsORC5's arginine fingers (or pistons), there is an arginine (R98) that could be interacting with the nucleotide bound to ORC5. An ORC5 Walker-B aspartate (D125) as well as ORC3-R98 were mutated in the context of HsORC1-5, and both are only slightly reduced for ATPase activity compared with wild-type activity (*Figure 3d*). This indicates that the ORC5-ORC3 interface does not significantly contribute to the observed ATPase activity of the HsORC1-5 complex.

## The missing puzzle piece – HsCDC6 can complete the ring

HsCDC6 binds to the core of HsORC as a second step in the assembly of the pre-RC (*Wu et al., 2014*). It is also an AAA+ ATPase with 29% sequence identity to ORC1, and in budding yeast, it is an active ATPase (*Randell et al., 2006*; *Speck and Stillman, 2007*). Both yeast and human ORC1 and CDC6 interact with each other (*Speck et al., 2005*; *Speck and Stillman, 2007*; *Hossain and Stillman, 2016*). We posit that CDC6 binds ORC1 with the same type of repeat arrangement as ORC1, ORC4 and ORC5 bind each other. We therefore built a homology model of CDC6 based on our structure of ORC1 and placed it with respect to ORC1 in the same relative arrangement as ORC1 is to ORC4 along with a putative ATP nucleotide between the two subunits in the same

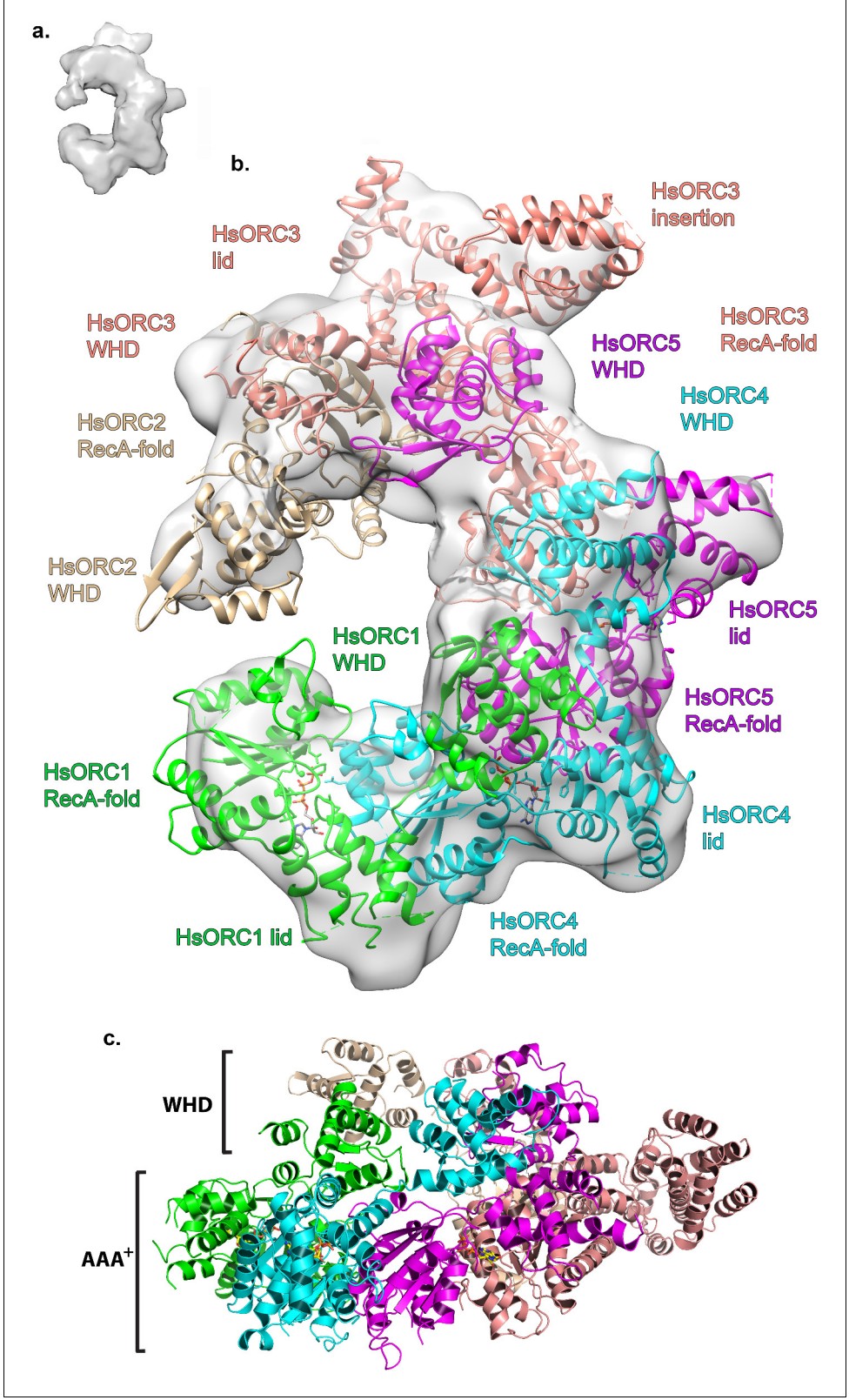

**Figure 4.** Structure of HsORC. (a) Cryo-EM density of HsORC1-5, back view. Note that the top and bottom are flipped compared with the maps shown in the supplementary Figures. (b) Ribbon diagram of HsORC modeled into cryo-EM density. The ORC motor module is colored as in *Figure 1*. ORC2 is shown in wheat and ORC3 is shown in salmon. ATP is shown in stick. (c) Side view with the WHD layer on top and the AAA+ layer on the bottom. The ORC3 insertion can be seen extending from the particle on the right.

*Figure 4 continued on next page*

*Figure 4 continued*

The following figure supplements are available for figure 4:

**Figure supplement 1.** Electron-density maps for the HsORC2/3 complex.

**Figure supplement 2.** Negative stain EM of the truncated human ORC1-5.

**Figure supplement 3.** Negative stain 3D EM map of the truncated HsORC.

**Figure supplement 4.** Generation of templates for automatic particle picking.

**Figure supplement 5.** 3D classification procedure used to derive the final 3D-density map of the HsORC.

**Figure supplement 6.** Eulerian angle distribution of the cryo-EM particles used in 3D refinement.

**Figure supplement 7.** Cryo-EM of the truncated human ORC1-5 complex.

**Figure supplement 8.** Structure of HsORC.

manner (*Figure 6a,b*, *Figure 6—figure supplement 1*). CDC6 fits within the HsORC model as if it were the missing piece in a puzzle. It completes the ring structure with its RecA domain between those of ORC1 and ORC2 on one layer and its WHD between those of ORC1 and ORC2 on the WHD layer. The ATP-binding site that is formed has all the characteristic residues in the proper positions. The CDC6 lid residue R388 is in the equivalent position as ORC1-R720 to act as the trigger. The helix of ORC1 that includes residues 661–674 is disordered in our structure, but by homology to ORC4, they would contain the R-finger as R666 and the tether as R670 in the proper position. Furthermore, the ORC1-Y634 discussed above points directly to the CDC6 Walker-B motif, specifically to CDC6-E285.

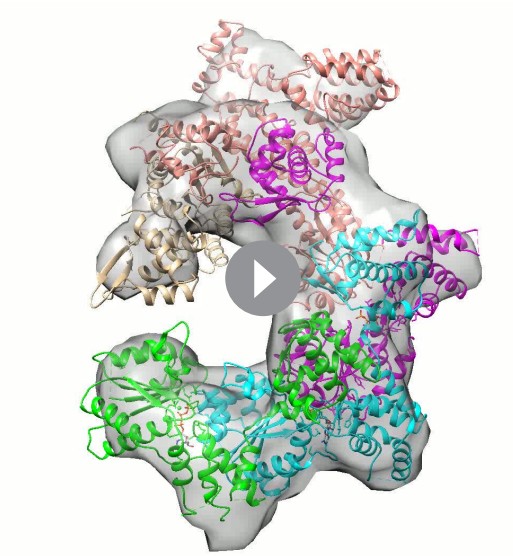

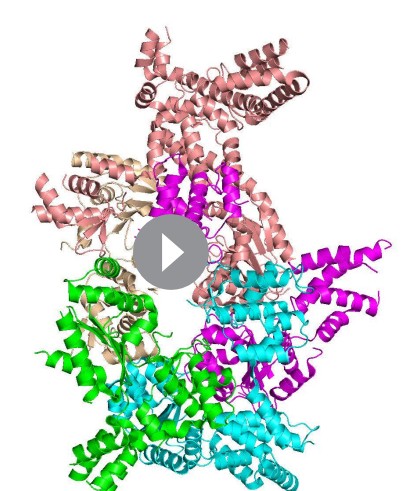

**Video 1.** Ribbon diagram of HsORC modeled into cryo-EM density. ORC1 is shown in green, ORC4 in cyan, ORC5 in purple, ORC2 in wheat and ORC3 in salmon. ATP is shown in stick.

**Video 2.** A morphing of the structure of DmORC into HsORC. HsORC1 moves to form an active ATP interface with HsORC4. HsORC2 moves out and up from the more 'collapsed' placement in DmORC.

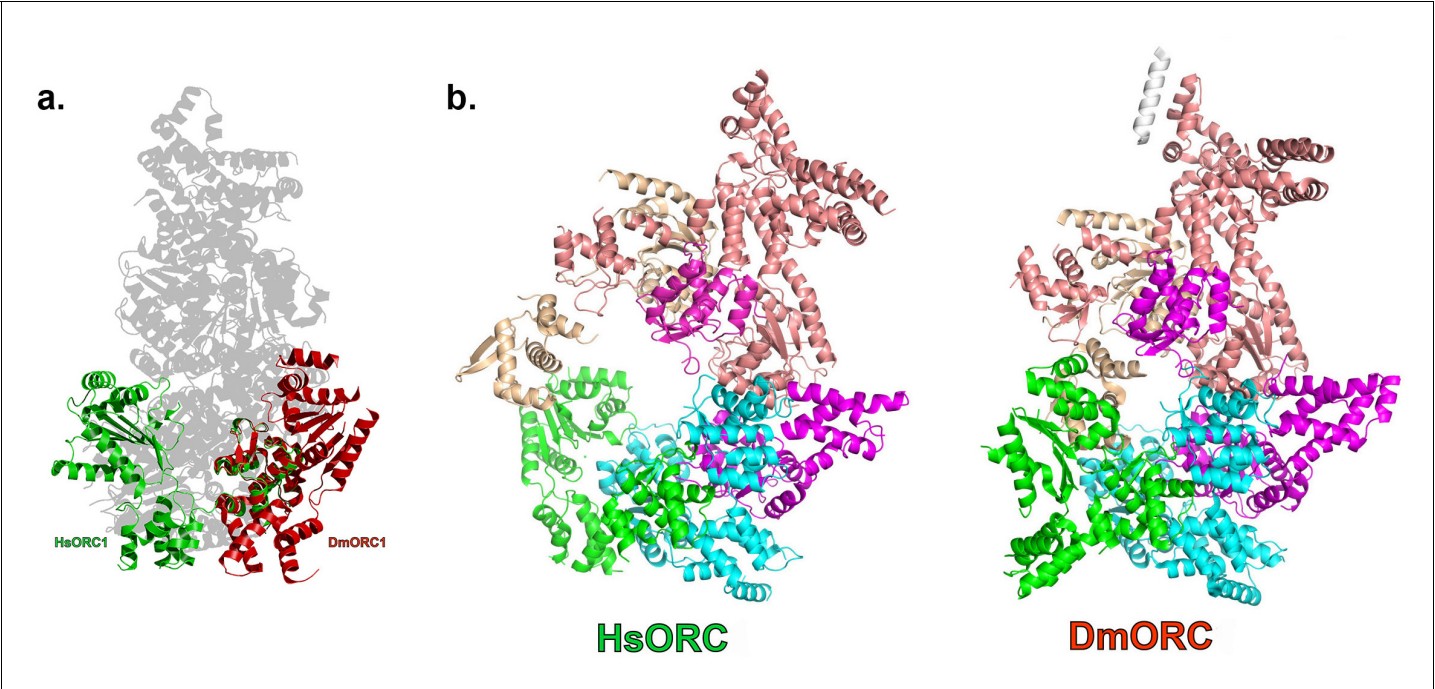

**Figure 5.** HsORC is in an active conformation. (**a**) Different positions of HsORC1 and DmORC1 in the context of the bigger particle. HsORC1 is shown in green, DmORC1 is shown in red and the rest of the HsORC complex is shown in grey. (**b**) The subunit organization in HsORC results in a more open architecture compared with DmORC.

## The DNA-binding site of HsORC

With the remarkable structural homology of the motor module to the clamp loaders in mind, we used the T4 bacteriophage clamp loader (T4-gp44) structure bound to an open clamp and a partially single-stranded, partially double-stranded primer template DNA (*Kelch et al., 2011*) to model the position of a double-stranded DNA bound to HsORC. This is one of the few structures of clamp loaders with DNA bound and was close in the relative orientations of the subunits to that of HsORC. We superimposed chains B, C and D of gp44 onto HsORC1/4/5. We used the resulting position of the DNA bound to gp44 and superimposed a longer canonical B-form, double-stranded DNA. The three RecA-folds of the AAA+ domains in ORC1, ORC4 and ORC5 wrap beautifully around the DNA double helix, tracking the minor groove of DNA and in the context of HsORC1-5, so does the RecA-like domain of ORC3 (*Figure 6c*, *Figure 6—figure supplement 2*). The WHDs contact the DNA seemingly through the major groove. The channel is wider than a snug fit to the DNA all around, although we expect (*Sun et al., 2013*) it to be in closer proximity to the active ATPase motor module elements.

## ATP hydrolysis and establishing the preRC

In the yeast system, ORC-Cdc6 loads the first Mcm2-7 hexamer, and the incoming Mcm3 C-terminus stimulates the ORC-Cdc6 ATPase activity (*Frigola et al., 2013*). In the absence of ATP hydrolysis, ORC-Cdc6 recruits the first Mcm2-7 hexamer that is bound to Cdt1, but the hexamer bound to the double-stranded DNA is not salt resistant (*Sun et al., 2013*; *Duzdevich et al., 2015*). Thus, ATPase activity is required for loading the Mcm2-7 hexamer onto DNA, resulting in a salt-resistant complex. Cdc6 is then released followed by the release of Cdt1 (*Ticau et al., 2015*). Another Cdc6 subunit is then involved in the recruitment of a second Cdt1-bound Mcm2-7 hexamer. ATP hydrolysis is linked to the removal of the second Cdc6 and the subsequent removal of Cdt1 and ORC from the origin (*Ticau et al., 2015*). Analogous to the clamp loader ATPases, we suggest that ATP hydrolysis by ORC-CDC6 would push down on the first MCM2-7 hexamer to trigger the Cdt1 release and allow recruitment of the second MCM2-7 – Cdt1 complex.

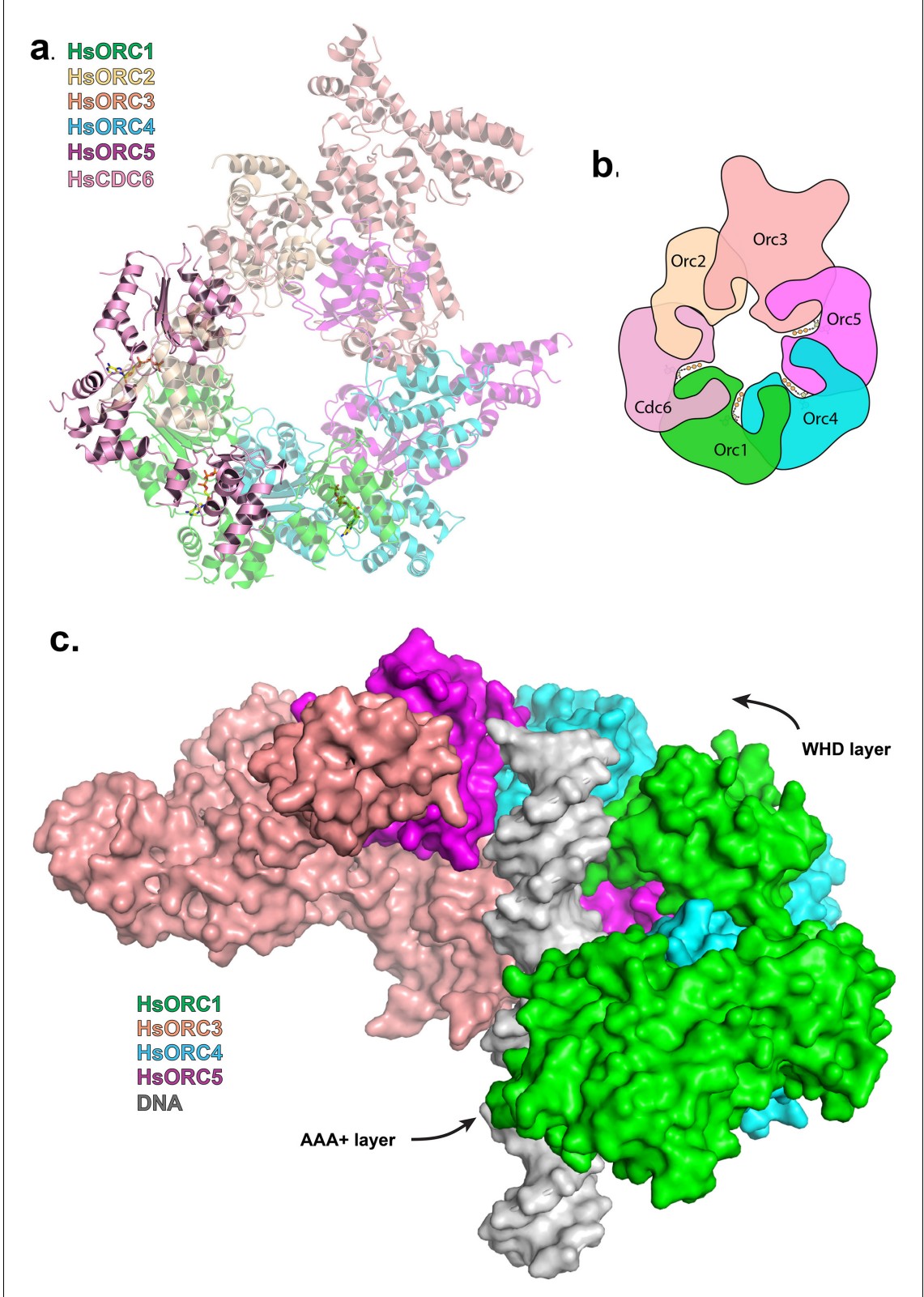

**Figure 6.** Modeled interaction of HsORC with CDC6 and DNA. (**a**) Model of HsORC-CDC6 with CDC6 in pink. HsORC1-5 subunits are shown in a transparent rendering. (**b**) Cartoon depiction of HsORC-CDC6 showing the organization of the particle with the smaller WHDs forming one layer on top of the bigger AAA+ layer on the bottom. It also clearly illustrates the WHD from one subunit sitting on top of the RecA-fold domain of the neighboring subunit. (**c**) Model of HsORC with DNA. ORC2 and CDC6 were removed for clarity.

*Figure 6 continued on next page*

*Figure 6 continued*

The following figure supplements are available for figure 6:

**Figure supplement 1.** Model of interaction of HsORC with CDC6.

**Figure supplement 2.** Model of interaction of HsORC with DNA.

## Concluding remarks

Although the structure of HsORC is substantially different from the known DmORC structure, it is very similar to the cryoEM structure of ORC from *Saccharomyces cerevisiae* (ScORC) observed in the context of the ORC-Cdc6-Cdt1-Mcm2-7 (OCCM) complex (*Sun et al., 2014*; Yuan, Riera, Bai, Sun, Nandi, Spanos, Chen, Barbon, Rappsilber, Stillman, Speck, and Li, personal communication). The structure of the yeast ORC-CDC6 complex determined by cryoEM is also similar to the structure we determined for human ORC and the model for human ORC-CDC6. Both HsORC and ScORC are in an active, ATPase state ready to function as a molecular motor, whereas the DmORC is in an inactive state that may be used for storage of these particles in the egg (*Bleichert et al., 2015*). The similarity of the human and yeast protein-loading machines is remarkable despite the fact that they interact with DNA in fundamentally different ways and that the yeast protein is stable during the cell division cycle, whereas the human complex is dynamically assembled and disassembled. Both are also highly similar to the DNA polymerase clamp-loading ATPases, showing that ATP-driven molecular machines that load ring-shaped proteins onto DNA have been re-purposed for multiple stages of DNA replication.

## Materials and methods

### Protein preparation

Condon optimized human origin recognition complex (HsORC) synthetic genes [NP_004144.2 HsORC subunit 1 (HsORC1), NP_006181.1 HsORC subunit 2 (HsORC2), NP_862820.1 HsORC subunit 3 (HsORC3), NP_859525.1 HsORC subunit 4 (HsORC4), NP_002544.1 HsORC subunit 5 (HsORC5)] were cloned into MultiBac baculovirus expression system (*Bieniossek et al., 2008*).

A series of truncations was designed to facilitate purification and monodispersity of human ORC. Trimming included an N-terminal truncation of ORC1 that removed the BAH domain, previously known to be rather flexible, and an N-terminal truncation of ORC2. ORC6 was also not included in the final complex used for our structural and biochemical analysis (*Figure 1—figure supplement 1*).

An initial 3D reconstruction of a truncated HsORC (ORC1-5) particle from single-particle negative-stain electron microscopy (EM) showed the particle to have a bilobal architecture. In addition, using a shorter construct of ORC5, we obtained an ORC subcomplex consisting of ORC1, ORC4 and ORC5. We therefore decided to separate the particle into the two lobes for crystallographic studies, one consisting of the N-terminally truncated ORC1, full-length ORC4 and a C-terminal truncation of ORC5. The other lobe consists of an N-terminally truncated ORC2 and full-length ORC3.

For recombinant expression of the heteropentameric HsORC1-5 gene-coding sequences were cloned into the pFL vector (pH promoter HsORC1 residues 471–861), pUCDM vector (pH promoter HsORC4 residues 1–436 (full-length), p10 promoter HsORC3 residues 1–712 (full-length) and pSPL vector (pH promoter HsORC2 residues 1–577 (full-length), p10 promoter HsORC5 residues 1–435 (full-length)). The Strep-Sumo-tagged ORC1 version of the complex was used for EM studies.

The motor module subcomplex (HsORC1/4/5) was cloned into pFL vector (pH promoter HsORC1 residues 471–861), pSPL vector (pH promoter HsORC4 residues 1–436 (full-length), p10 promoter HsORC5 residues 1–284). Finally, the expression cassette for the complex of HsORC2 and HsORC3 (HsORC2/3) was cloned into pFL vector (pH promoter HsORC2 residues 231–577) and pUCDM vector [p10 promoter HsORC3 residues 1–712 (full-length)].

A two-Strep-SumoStar (TSS) tag (based on the One-Strep-SumoStar [OSS] tag [*Schalch et al., 2011*]) was added at the N-terminus of HsORC1 to facilitate affinity purification of HsORC1-5 and HsORC1/4/5, while in the HsORC2/3 case the TSS tag was N terminally fused in frame with HsORC2.

Sf9 insect cells were infected with baculovirus in Hyclone CCM3 media (GE Healthcare Life Sciences, Pittsburgh, PA) for 48 hr.

For selenomethionine protein expression, 1 l of High-five cells at density of $2*10^6$/ml were diluted at 1:2 ratio using ESF921 methionine-deficient media (Expression systems, Davis, CA) 2 days prior to baculovirus infection. The cells were diluted again the following day using the same media and were allowed to grow for an additional day until reaching density of $2*10^6$/ml (total of 4 l). The cells were then infected with 30 ml virus per liter of High-five cells. After 4 hr, the cells were centrifuged for 5 min at 800 g and washed in 100% methionine-deficient media, centrifuged again and resuspended in methionine-deficient media. After 1 hr, Selenomethionine (Sigma-Aldrich, St. Louis, MO) was added to a final concentration of 198 mg/l. The cells were harvested 48 hr later.

All purification steps were performed at 4°C. Cell pellets were thawed on ice and resuspended in lysis buffer (50 mM HEPES-NaOH (pH 7.5), 200 mM KCl, 5 mM dithiothreitol (DTT), 5 mM adenosine triphosphate (ATP), 10 mM monopotassium glutamate (KGlu)). Following sonication, the lysate was centrifuged for 60 min at 100,000 g and the supernatant was loaded onto 5 ml of StrepTactin Superflow resin (IBA, Goettingen, Germany). The beads were washed and recombinant HsORC1-5 complex was eluted with lysis buffer, which contained 25 mM desthiobiotin. The protein complex was then further loaded onto a HS20 cation-exchange column (POROS, ThermoFisher Scientific, Waltham, MA). The column was washed with buffer A (50 mM HEPES-NaOH (pH 7.5), 100 mM KCl, 5 mM DTT, 5 mM ATP, 10 mM KGlu), and eluted with a 30 min, 0.5 ml/min gradient of buffer B (50 mM HEPES-NaOH (pH 7.5), 1M KCl, 5 mM DTT, 5 mM ATP, 10 mM KGlu). Peak fractions eluting at ~350 mM KCl were concentrated and loaded onto a Superdex 200 Increase 10/300 GL size exclusion column (GE Healthcare Life Sciences, Pittsburgh, PA) equilibrated with buffer A.

Recombinant HsORC1/4/5 complex was purified similarly to HsORC1-5, with the following differences: the TSS tag was cleaved with Sumo* protease at 4°C on the StrepTactin Superflow resin (IBA, Goettingen, Germany). The elution fraction was diluted twofold into buffer A and loaded onto a HQ20 anion-exchange column (POROS, ThermoFisher Scientific, Waltham, MA). A major peak containing all three subunits eluting at ~0.2 M KCl was pooled and was loaded onto a size exclusion Superdex 200 Increase 10/300 GL column (GE Healthcare Life Sciences, Pittsburgh, PA) equilibrated with buffer A. In both cases, ATP (5 mM) was added to all buffers throughout the purification procedure in order to maintain complex integrity; however, $Mg^{2+}$ was excluded to avoid ATP hydrolysis. The cleavage efficiency and purity was verified by SDS-PAGE.

The HsORC2/3 complex was purified in two steps by StrepTactin affinity chromatography, cleaved on the beads with Sumo* protease at 4°C and followed by size exclusion chromatography using a Superdex 200 Increase 10/300 GL column (GE Healthcare Life Sciences, Pittsburgh, PA) equilibrated with buffer A. The integrity of the HsORC2/3 complex was not affected in the presence or absence of ATP.

## Protein crystallization and data collection

Crystals of the HsORC1/4/5 motor module were obtained by hanging-drop vapor diffusion by mixing HsORC1/4/5 at 10 mg/ml with an equal volume of a buffer of 22% ethylene glycol and 8% propylene glycol. Crystals appeared overnight at 17°C, were harvested quickly without additional cyroprotectant, and flash frozen by plunging into liquid nitrogen. We should note that crystals disappear within 30 hr if not harvested, possibly due to slow ATP hydrolysis in the crystals.

X-ray diffraction data were collected to 3.4 Å resolution at beamline ID19 at the Argonne National Laboratory (APS). Diffraction data were indexed, integrated and scaled using XDS (*Kabsch, 1993*) and aP_scale (*Vonrhein et al., 2011*), respectively, via the AUTOPROC (*Vonrhein et al., 2011*). The structure of HsORC1/4/5 was determined by molecular replacement with PHASER (*McCoy et al., 2007*) using *Drosophila* ORC (DmORC) PDB entry 4XGC (DmORC1 residues A820-A919, DmORC4 residues D5-D457 and DmORC5 residues E1-E182) (*Bleichert et al., 2015*) as search model. It crystallized with two independent complexes in the asymmetric unit. Initial MR phases were further improved by twofold averaging, solvent flattening and automated model rebuilding as implemented in PHENIX (mr_rosetta) (*Terwilliger et al., 2012*), resulting in a partially refined model that was completed with manual correction using COOT (*Emsley et al., 2010*). The anomalous differences for the SeMet peak data confirmed the sequence register and accuracy of the model. The geometry of the refined model was further improved using PHENIX (rosetta.refine) (*DiMaio et al., 2013*) and the inclusion of riding hydrogens. The six ATP molecules were modeled

into the σA-weighted Fo-Fc difference electron-density map (*Figure 1—figure supplement 2*) toward the end of protein refinement. The final model was refined to an $R_{work}/R_{free}$ of 0.242/0.281 by using two-fold noncrystallographic symmetry restraints, TLS and isotropic B-factor refinement as implemented in PHENIX (*Adams et al., 2010*). It contains two HsORC1/4/5 particles in the asymmetric unit, six ATP molecules and four $Mg^{2+}$ ions. The final model was validated with MolProbity (*Davis et al., 2007*), with no Ramachandran outliers (*Table 1*).

The two complexes are very similar with a RMSD of 0.53 Å on Cα's. The copy of the motor module represented by chain A (HsORC1), chain D (HsORC4) and chain E (HsORC5) had somewhat better electron density and was used for all structural analysis described in the manuscript.

Crystals of HsORC2/3 were obtained by hanging-drop vapor diffusion after mixing protein at 10 mg/ml with an equal volume of 9.5% PEG20000, 50 mM tri-Sodium-citrate, 60 mM Citric Acid. These crystals typically grew overnight as thin plates, with the thinnest dimension rarely exceeding 0.01 mm. They belong to monoclinic space group $P2_1$ with four HsORC2/3 particles in the asymmetric unit and diffracted to 6 Å resolution. The crystals were found to possess pseudo merohedral twinning as well as translational non-crystallographic symmetry.

Nonetheless, a molecular replacement solution was obtained using a homology model for HsORC2 and HsORC3 built from the *Drosophila* ORC2 and ORC3 structures (PDB ID: 4XGC chains B residues 326 to 508 and C residues 46 to 721) by Sculptor and that was further energy minimized in Rosetta. The correct molecular replacement solution was obtained using PHASER-MR with four HsORC2/3 molecules placed in asymmetric unit by taking into account translational NCS. Domain adjustments against a four-fold NCS averaged Phaser 2Fo-Fc sigma weighted map were done by iterative application of the PHENIX model_morph procedure.

The model was subsequently refined against pseudo-merohedrally twinned data to a resolution of 6 Å as implemented in PHENIX. The twinning operator (h,-k, -l) and a twin fraction of 0.35 together with NCS, Ramachandran and secondary structure restraints. The refinement procedure produced a clear electron-density map to a resolution of about 6 Å, which showed α-helical features but had insufficient resolution for side-chain rebuilding (*Figure 4—figure supplement 1*) (*Table 2*). The WHD of ORC2 could not be unequivocally placed in this electron-density map. From packing considerations, it could assume the same position as in the fly complex only in two of the complexes, but this would cause significant clashes in the other two molecules. Although we do not consider this to be a well-refined crystal structure of the HsORC2/3 subcomplex, we do consider this to be a good docking model for the cryoEM structure (see below).

## Cryo-electron microscopy
### Sample preparation and data collection
HsORC1-5 was diluted to ~0.6 mg/ml in 0.5 mM ATPγS and 20 mM magnesium acetate buffer. Negative-stain electron microscopy was used to confirm the sample homogeneity and to determine an initial 3D reconstruction at ~20 Å by using a low-pass filtered (55 Å) ScORC EM map as the starting model (*Figure 4—figure supplement 2 and 3*). For cryoEM grid preparation, we applied 2.5 μl of HsORC1-5 at a final concentration of 0.6 mg/ml to a glow-discharged C-flat 1.2/1/3 holey carbon grid, incubated for 10 s at 6°C and 90% humidity, blotted for 3.5 s, and then plunged the grid into liquid ethane using an FEI Vitrobot IV (Hillsboro, OR). The grids were loaded into an FEI Titian Krios electron microscope operated at a high tension of 300 kV and images were collected semi-automatically with SerialEM (*Mastronarde, 2005*) under low-dose mode at a magnification of ×29,000 and a pixel size of 1.01 Å, with an under-focus range of 1.5 to 3.5 μm. A Gatan (Pleasanton, CA) K2 summit direct electron detector was used under super-resolution mode for image recording. The dose rate was 10 electrons per pixel per second with 5 s total exposure time. The total dose was divided into a 25-frame movie with a 0.2 s exposure per frame.

## Image processing and 3D reconstruction
2765 raw movie micrographs were collected (*Figure 4—figure supplement 4*). The movie frames were first aligned and superimposed using the Motioncorr program (*Li et al., 2013*). Contrast transfer function parameters of each aligned micrograph were calculated using the program CTFFIND4 (*Rohou and Grigorieff, 2015*). All remaining steps, including semiautomatic particle picking, 2D classification, 3D classification, 3D refinement, and density map post-processing were performed

using Relion-1.4 (*Scheres, 2015*). Many particles appeared to be disassociated HsORC1/4/5 and HsORC2/3 subcomplexes. Guided by several published homologous ORC structures from yeast and Drosophila, we were able to manually select intact particles. We manually picked ~3000 particles in different views to generate initial 2D averages, which were used as templates for subsequent automatic particle selection (*Figure 4—figure supplement 4*). Automatic particle selection was then performed for the entire data set. 532,782 particles were initially selected. The particles were then carefully examined and 'bad' particles were removed. 2D classifications of all remaining particles were performed (*Figure 4—figure supplement 4c*) and particles in unrecognizable or partial classes by visual inspection were removed. Particles were then sorted by similarity to the 2D references; the 10% of particles with the lowest z-scores were deleted from the particle pool. A total of 100,543 particles were used for 3D classification (*Figure 4—figure supplement 5*). We derived six 3D models from the dataset and found only one model suitable for further refinement; the other five models were distorted or partial structures and those particles were discarded, leading to a dataset size of 10,357 particles. This selection was aided by significant prior knowledge of the general shape and size of the ORC structure. Because of the small dataset size and somewhat uneven angular distribution, we took a conservative approach and binned the particle images by a factor of 4, leading to a final sampling of 4.04 Å per pixel. The binned dataset was used for further 3D refinement, resulting in the described 3D-density map. The resolution of the map was estimated to be 18 Å by the so-called gold standard Fourier shell correlation, at a correlation cutoff of 0.143. However, the FSC curve started to drop off at a resolution as low as 30 Å, likely due to uneven angular distribution (*Figure 4—figure supplement 6*), suggesting that the map may only contain reliable information up to ~20 Å resolution (*Figure 4—figure supplement 7*). The 3D-density maps were corrected for the detector modulation transfer function and sharpened by applying a negative B-factor of −479 Å$^2$. Additional statistics are shown in *Table 3*. The final 3D map was used rigid body docking of the HsORC1/4/5 and HsORC2/3 structures.

## Model building

The HsORC motor module was easily docked into the EM reconstruction density map with Chimera (*Pettersen et al., 2004*). The HsORC2/3 model without the HsORC2 WHD was then docked into the map, followed by a homology model of the WHD of HsORC5 built using the fly homolog. A blob of density emanating from the RecA-fold domain of ORC2 was a clear indication for the positioning of the WHD of HsORC2, using the available structure (PDB code 5C8H). Refinement was done by keeping the motor module ORC1/4/5, the RecA fold of ORC2, the complete ORC3, the WHD of ORC5 and the WHD of ORC2 as discrete rigid bodies. Flexible fitting resulted in reorientation of ORC subunits without biological justification, and therefore not used.

**Table 3.** Cryo-EM data collection and refinement statistics of HsORC1-5.

| Data Collection | |
| --- | --- |
| EM equipment | FEI Titan Krios |
| Voltage (kV) | 300 |
| Detector | Gatan K2 |
| Pixel size (Å) | 1.01 |
| Electron dose (e-/Å2) | 50 |
| Defocus range (μm) | −1.5 ~ −3.5 |
| Reconstruction | |
| Software | RELION 1.4 |
| Number of final particles | 10,357 |
| Resolution (Å) | 20 |
| Map sharpening B-factor (Å$^2$) | −479 |

## ORC ATPase assays

Reactions (20 µl) were typically carried out in buffer containing 50 mM Hepes-NaOH (pH 7.5), 200 mM KCl, 5 mM MgCl$_2$, 5 mM DTT, 50 µM cold ATP, 5 µCi [γ -$^{32}$P] ATP (PerkinElmer, Waltham, MA) and 10 µM HsORC1/4/5 or HsORC1-5 complex as indicated (unless stated otherwise). Reactions were incubated at 37°C for 30 min and quenched with 2 µl of 500 mM EDTA (pH 8.0). 2 µl of each reaction was then spotted on a PEI-cellulose TLC plate (Selecto Scientific, Suwanee, GA), developed using 0.8 M LiCl and 0.8 M acetic acid as solvent, and quantified on a Phosphorimager (GE Health-care Life Sciences, Pittsburgh, PA). For kinetic analyses, 10 µM HsORC was incubated with a titration of total ATP concentrations (cold and radioactive) of 12.5–200 µM in the aforementioned buffer. Reaction rate remained linear at least for 30 min (data not shown), and the ATP hydrolysis rate was used as the initial velocity for kinetic analyses. Initial velocity was plotted against total ATP concentrations for Michaelis-Menten equation fitting (GraphPad Prism, La Jolla, CA).

Figures were generated by using PyMol (The PyMOL Molecular Graphics System, Version 1.8 Schrödinger, Cambridge, MA).

## Accession codes

Coordinates and structure factors for HsORC1/4/5 (PDB ID 5UJ7) and HsORC2/3 (PDB ID 5UJ8) have been deposited to the Protein Data Bank. Note that HsORC2/3 is at low resolution, but suitable as a docking model for the cryoEM 3D map. The cryoEM-density map (EMD-8541) has been deposited to the Electron Microscopy Data Bank, and the coordinates for the docked HsORC1-5 model has been deposited to the Protein Data Bank (PDB ID 5UJM).

# Acknowledgements

We thank members of the Joshua-Tor laboratory for helpful comments and suggestions. We thank Jonathan Ipsaro for his artistry in producing *Figure 6b*. We thank Stephen Ginell and Boguslaw Nocek (SBC) and Raj Rajashankar, David Neau and Jonathan Schuermann (NE-CAT) at the Advanced Light Source, Argonne National Laboratory. We thank Zhiheng Yu, Chuan Hong and Rick Huang at Janelia Research Campus of HHMI for assistance with cryoEM data collection. We thank the CSHL Mass Spectrometry Shared Resource which is supported by Cancer Center Support Grant 5P30CA045508. This work was supported by NIH grants R01-GM111742 (to HL),R01-GM45436 and P01-CA13016 (to BS), the Robertson Research Fund of Cold Spring Harbor Laboratory and the Cold Spring Harbor Laboratory Women in Science Award (to LJ). LJ is an investigator of the Howard Hughes Medical Institute.

# Additional information

### Funding

| Funder | Grant reference number | Author |
| --- | --- | --- |
| National Institute of General Medical Sciences | R01-GM111742 | Huilin Li |
| National Cancer Institute | P01-CA13016 | Bruce Stillman |
| Howard Hughes Medical Institute | | Leemor Joshua-Tor |
| National Institute of General Medical Sciences | R01-GM45436 | Bruce Stillman |

The funders had no role in study design, data collection and interpretation, or the decision to submit the work for publication.

### Author contributions

AT, Conceptualization, Data curation, Formal analysis, Validation, Investigation, Visualization, Methodology, Writing—original draft; KFO, Conceptualization, Data curation, Formal analysis, Validation, Visualization, Methodology, Writing—original draft; ZY, Data curation, Formal analysis, Validation, Visualization; JS, Data curation, Formal analysis, Validation, Visualization, Methodology; EE, Data

curation, Formal analysis; HL, Conceptualization, Data curation, Formal analysis, Supervision, Validation, Visualization, Methodology, Writing—original draft, Writing—review and editing; BS, Conceptualization, Funding acquisition, Writing—original draft, Project administration, Writing—review and editing; LJ, Conceptualization, Data curation, Supervision, Funding acquisition, Visualization, Methodology, Writing—original draft, Project administration, Writing—review and editing

### Author ORCIDs
Bruce Stillman, http://orcid.org/0000-0002-9453-4091
Leemor Joshua-Tor, http://orcid.org/0000-0001-8185-8049

## Additional files

### Major datasets
The following datasets were generated:

| Author(s) | Year | Dataset title | Dataset URL | Database, license, and accessibility information |
|---|---|---|---|---|
| Ante Tocilj, Kin Fan On, Elad Elkayam, Leemor Joshua-Tor | 2017 | Human Origin Recognition Complex subunits 1, 4 and 5 | http://www.rcsb.org/pdb/search/structidSearch.do?structureId=5UJ7 | Publicly available at the RCSB Protein Data Bank (accession no: 5UJ7) |
| Ante Tocilj, Kin Fan On, Elad Elkayam, Leemor Joshua-Tor | 2017 | Human Origin Recognition Complex subunits 2 and 3 | http://www.rcsb.org/pdb/search/structidSearch.do?structureId=5UJ8 | Publicly available at the RCSB Protein Data Bank (accession no: 5UJ8) |
| Ante Tocilj, Leemor Joshua-Tor | 2017 | Structure of the active form of the human Origin Recognition Complex and its motor module | http://www.rcsb.org/pdb/search/structidSearch.do?structureId=5UJM | Publicly available at the RCSB Protein Data Bank (accession no: 5UJM) |
| Zuanning Yuan, Ante Tocilj, Jing-chuan Sun, Huilin Li, Leemor Joshua-Tor | 2017 | Structure of the active form of the human Origin Recognition Complex and its motor module | http://www.ebi.ac.uk/pdbe/entry/emdb/EMD-8541 | Publicly available at the EMDataBank (accession no: EMD-8541) |

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
