## [Decision Letter]

Thank you for submitting your article "Structure of the active form of human Origin Recognition Complex and its ATPase motor module" for consideration by *eLife*. Your article has been reviewed by John Kuriyan as the Senior and Reviewing Editor, three peer reviewers, and another Reviewing Editor with expertise in electron cryomicroscopy. As you will see below, the review has identified certain important issues that need to be addressed before the paper can be considered further.

Review:

This manuscript from Joshua-Tor and colleagues reports structural studies of the Origin Recognition Complex, combining crystal structures and cryoEM. The structure of active ORC has been sought for a long time.

Specifically, in this manuscript, Joshua-Tor and coworkers report the ~3.4 Å resolution crystal structure of the motor module of human origin recognition complex (HsORC), which consists of ORC1, ORC4 and ORC5 subunits, and ATPase-activity analysis of the three-subunit motor module containing disease-related mutations. To complement and confirm the crystal structure, the authors also report a cryo-EM structure of the five-subunit HsORC1-5 with the sixth subunit CDC6 missing. This follows work from the Berger lab published in Nature last year of the (almost) complete structure of the *Drosophila* Orc. The Berger structure was of the inactive (nucleotide free) state of the complex and it also lacked the N-terminal regions of Orc1, Orc2 and Orc3 and the Orc6 TFIIB region. Similar deletions were used for these human ORC sub complexes in the present work (N-terminal deletions of Orc1 and Orc2 plus C-terminal deletion of Orc5). Orc6 is omitted entirely. Although the Berger crystal structure contained a more complete ORC complex from *Drosophila* (DmORC1-5), that structure represents a repressed state with Orc1 adopting an inactive domain arrangement. In contrast, the crystal structure of HsORC1/4/5 in the present work contains three ATPs bound at inter-subunit interfaces as expected, and the motor module is in an active form and similar to the structures of the clamp loader RFC from yeast, *E. coli* and T4 phage. The low resolution EM structure confirms the subunit arrangement revealed by the crystal structure and includes the two missing subunits ORC2 and ORC3. The ATPase activity assay and characterization of disease mutations further strengthen the structural results.

The reviewers recognize that the results reported here are exciting and lead to new hypotheses for ORC function/mechanism. Thus, in principle, the scope of this work makes it suitable for publication in *eLife*. Nevertheless, the reviewers have identified two sets of serious issues with the paper. One issue can, presumably, be dealt with through revision, but the other, concerning the cryoEM data, will require extensive reanalysis of data, or acquisition of new data. The reviewers and the editors feel that the EM analysis is essential to the conclusions of the paper, but that it will have to be significantly improved, augmented and validated before this work can be considered for publication in *eLife*. The reviewers also note that for the EM to support the conclusions of the paper, a resolution of 10 Å is not necessary. A reliable reconstruction at lower resolution, say 20 Å, would be fine, provided that it is indeed reliable and validated.

It is unclear to the reviewers whether the work needed to improve and validate the EM analysis is feasible in the normal two-month time frame within which we seek revisions. If it is possible to resubmit this paper within a two-month window, then we would be pleased to consider a revised version. If not, the authors may wish to dedicate more time to fixing the essential aspects of the EM analysis and either submit a new paper to *eLife*, or submit the work elsewhere. Please let us know how you plan to proceed.

Essential issues to fix regarding the electron microscopy:

The reviewers are concerned that there is something seriously wrong with the EM analysis, as the reconstructions represented in the figures do not square with the statistics. The only figure shown for the cryoEM structure is Figure 3 which is patently not at 9.9 Å resolution – it appears to be more like 20 Å, at best. At 10 Å resolution hints of α-helices should be visible in the density. It is also puzzling that the 2D classes in the Figure 3 supplement show detail that would be consistent with a 10 Å structure. How do these 2D classes relate to the final model? How many particles are averaged here and are these the from the final 10,000? There needs to be considerably more information to assess this and the model they present especially with respect to the Berger model of the inactive form of the complex.

From the discussion in Methods it seems they started with 530,000 particles then discarded 80% of those. The remaining 100,000 particles were divided into 6 classes and they then chose a class with just 10,000 particles, which can't even have been the largest class. This means the cryoEM structure actually represents less than 2% of the primary particles. Consequently, there is little confidence as to what this subset might represent. If it were representative of a selected subset that were, for example, a different conformation, it would still require considerable care and biochemical backup to validate this selection. A concern is that the majority of the particles may be "distressed" particles that are missing subunits, etc. The image shown in the supplementary would support this view as there seems to be many small particles that are probably partial complexes. Under such conditions, one needs to take great care (and have significant justification) in selecting out such a small subset for detailed analysis without describing what the other particles might represent.

The reviewers recognize that it can be justifiable under some circumstances to use only 10k out of 100k particles, but the micrograph in Figure 3—figure supplement 1 Figure looks quite messy. If this is a 'representative' one, then the auto-picking could have picked up a lot of bad (clung together?) particles and only a small subset would be suitable for 3D reconstruction. Is this what happened? Have the authors considered the implications of this carefully?

The 2D classes do look more detailed, but apart from two very similar views of the complex, only some classes which show what looks like partial-complexes are shown. Perhaps the angular distribution is very uneven and no sound 3D map could actually be calculated? An angular distribution plot would answer that question.

The Fourier Shell Correlation (FSC) curve (Figure 3—figure supplement 1) is problematic. FSC curves of good cryoEM maps hover around 0.8 to 1 for most of the resolution range and then dip steeply near the resolution limit. This raises concerns regarding overfitting noise. The Methods do not mention anything about map validation. A stringent test would be to randomize all phases above 20 or 30 Å and then see if they come back during refinement. This will almost certainly show that the actual resolution is somewhere between 20 and 30 Å.

To summarize our position on the EM analysis:

There is a significant concern that the complex was either partially degraded (most likely) or that there were significant impurities in the preparation that gave rise to the mix of particle sizes. Given this, the cryoEM reconstruction needs to be carefully validated, and presented at an appropriate resolution. In reporting on their data after reanalysis, the authors should pay attention to the following points:

1) The authors should provide a display of all 2D classes, not just the best ones. The same is true for the 3D classes – all six should be shown.

2) The authors mention an initial study using negative stain. These data should be shown, including the 3D reconstruction they obtained from these data. Furthermore, they should explain how they obtained this first 3D reconstruction. Finally, was this 3D reconstruction used to initiate processing of the cryo-EM data using Relion? If not, which initial reference did they use?

3) The authors should show examples of the initial 3000 particles picked manually and the 2D class averages obtained from them that were used as templates for subsequent automatic picking. It is quite possible that this initial picking step biased the rest of the picking to favor particular views and exclude other views of the complex.

4) The authors should also explain what they think is "wrong" with the majority of particles that they call "bad".

5) The authors should show a random sampling of particles from the 10,000 particles that were used for the final 3D reconstruction, together with matching projections to show that the alignments were plausible and the 3D reconstruction is consistent with these particles.

6) In the present manuscript, the presentation of the cryo-EM structure of HsORC1-5 in Figure 3 indicates poor fitting of the ORC2 model to the density map, and there is no EM density map superimposed onto the structure in the orthogonal view shown in Figure 3. The authors need to describe how good or bad the fit is, once the EM reconstruction is validated.

Essential revisions concerning other aspects of the paper:

The reviewers are concerned that the paper does not place this work properly in the context of the previous paper from Berger on the structure of the *Drosophila* ORC complex. The Introduction should begin by providing the context, outlining what is known previously at a structural level. The authors do not discuss how their structure relates to MCM loading mechanism in sufficient detail. The architectural similarities to the sliding clamp loader are noted and would naturally lead to some hypotheses for how ORC loads MCM. However, the authors don't present possible mechanisms that their structure naturally points to.

More specifically, references to the Berger structure are scant and it seems the authors describe aspects of the structure that are the same or at least very similar, which should be acknowledged. The first concession that there is any similarity is in the subsection “HsORC1 exists in an active conformation”, when they only mention the Berger work to say the architecture of the complex is similar but the conformation is different, although they provide no figure for the comparison to be assessed. In fact, the Berger coordinates were used for molecular replacement, although the methods do not say whether they were used as a single unit in each case (Orc1, 4, 5 and Orc2, 3) or the subunits were found separately. We presume the latter, since the paper suggests the conformations are different although the only comparison of the structures is of one subunit in the panel in Figure 2. The cursory nature of the references to the Berger paper is inappropriate, when it is evident that the similarity is in fact significant.

The important difference is that the present work has crystal structures (of sub-complexes) that contain bound nucleotides. However, it is well established in these AAA systems that it is essential to have details of the strain induced around rings or spirals when nucleotides are bound, as this creates conformational changes that drive the biochemical processes. Clamp loading is one pertinent example since they draw several comparisons with that system. The explanation for now nucleotide binding induces the conformational changes observed is not well explained, and should merit additional discussion.

To summarize this aspect of the essential revisions, the paper would be substantially improved by explaining the system better to the general reader, and discussing the new findings in the context of the earlier structural results on the ORC complex, rather than (mainly) independently of them.

[Editors' note: further revisions were requested prior to acceptance, as described below.]

Thank you for resubmitting your work entitled "Structure of the active form of human Origin Recognition Complex and its ATPase motor module" for further consideration at *eLife*. Your article has been favorably evaluated by John Kuriyan (Senior and Reviewing Editor) and four reviewers, one of whom, Niko Grigorieff, is a member of our Board of Reviewing Editors with expertise in cryoEM.

We retain our opinion that your paper presents an important set of results. Nevertheless, based on a discussion between the reviewers and the editors, we have concluded that the paper is not yet at a stage where we are prepared to accept it at *eLife*. Although we usually do not go beyond one round of review, there are two issues, one major and one less critical, that cause us to return the manuscript for further revision. The major issue, as before, concerns the integrity of the electron microscopy analysis. The other issue concerns the quality of the narrative in the manuscript and the illustrations, and should be easily fixed with some attention to clarity and detail. As a reminder, *eLife* has no page limits, allowing authors to expand at will and to the extent they deem appropriate.

Electron Microscopy:

The authors were able to clarify/answer a few points raised in the original review:

They now show more than six 2D class averages, although they still do not show all the class averages they obtained. Neither do they say how many they obtained.They also show more of the original data, including data from their negative-stain study and from the initial 3000 particles and resulting 2D class averages that they used as templates for particle picking.They also show all six 3D classes obtained from the 100k particles selected based on the 2D classification.They included an angular plot showing the determined orientations for the final 10k particles. This plot is interesting because it suggests that the problem of preferred orientation was not as severe as one might expect from the 2D class averages (which show only a few views). This means that there is a discrepancy between the plot and the 2D class averages.The authors also concede that the cryo data suggests that the complex was falling apart, leading to significant heterogeneity.The original claim of 10 Å resolution was toned down to ~14 Å. This is going in the right direction but maybe 20 Å might be even more appropriate.

Some of the original problems remain, including the strange-looking FSC curve, the substantial heterogeneity of the selected particles, and the extremely small percentage (2%) of data that ended up in the final reconstruction. The negative-stain data actually look quite reasonable, and we wonder if the 3D reconstruction they obtained from them would have been of better quality than the cryo-EM structure. It seems that the main problem is that the complex is falling apart during the freezing process (as seen also with many other complexes). This would explain the heterogeneity, the fact that only 2% of the particles can be merged into a 3D structure and that many particles are misaligned and thus give rise to the decent-looking angular distribution even though the 2D class averages show only a few recognizable views. The likely remaining heterogeneity and misalignment in the final 10k particles may also explain why the FSC looks so unusual. If some of the broken pieces of the complex can still be aligned to the relevant parts of the whole complex, they may still contribute some signal in the final reconstruction.

Conclusion: We believe that the cryo-EM work, as presented, has flaws and is of low quality. However, given the negative-stain structure and assuming that there is some truth in the cryo-EM structure at ~20 Å resolution, the data, processed appropriately to lower resolution, might be sufficient for what the authors want to show: activation of ORC results in an enlarged cavity for DNA and that there is a gap in the AAA ring that can be filled by ORC6.

Alternatively, given that the negative-stain reconstruction is actually quite good compared to the cryo-EM reconstruction, the authors may wish to include that either in the main analysis or it in the deposition of structures in the database.

The revised manuscript should include in the main text a discussion of the EM issues (e.g., those highlighted above) and the approaches taken to include a reliable interpretation in the final analysis. The discussion should be such that an interested reader is left with a good sense as to the extent to which the reconstruction should be trusted. It is essential that the reviewers are reassured on these points.

Manuscript text and figures:

We feel that the manuscript, as written and illustrated, falls short of presenting the work in a way that properly reflects the importance of the results. The current manuscript risks losing the attention of readers who are not well versed in the topic.

Please consider the following points seriously in preparing a revised version:

1) The Introduction is striking for its brevity, and it fails to set the stage for the general reader to appreciate the results. It consists of two paragraphs. The first is a very long run-on paragraph in which many concepts are elided. The second paragraph describes the crystal structures, but provides no introduction to the EM analysis. The Introduction should be rewritten with an eye to clarity and also to maximizing appreciation of the results.

2) The figures could be improved substantially. The views of the structures could be augmented by showing an alternative orientation in which the nucleotide coordination is emphasized rather than the overall assembly architecture. The labeling of the figures is sparse, causing the reader to strain to understand all the components. We have no set limits – take the space that you feel is appropriate to illustrate your results to best effect.

3) In comparing the structure obtained here to other structures, it is very difficult to judge changes in inter-subunit orientation. For example, in Figure 1, there is no superposition of ORC with RFC to show that the inter-subunit orientations are preserved. Likewise, one cannot readily tell that the inter-subunit orientations are preserved within ORC itself.

4) The revised manuscript mentions the crystal structure of DmORC in the Introduction, but categorically labels it as "inactive" without giving any explanation as to how "inactive" is defined or describing what the structure includes. (We appreciate what you mean, but we fear that the general reader would not.) Following a too brief discussion of the DmORC structure, the authors state that "we report the structure of human ORC in a functionally active state". But their crystal structure is of a three-subunit sub-complex and not the six-subunit human ORC. The "activity" refers to only ATP hydrolysis by the sub-complex and not initiation of DNA replication. Some explanatory text could tie this all together and present the new work in the best light.

Additional important points:

1) In the second paragraph of the subsection “The nucleotide binding sites”, the authors discuss ORC5-R104 as the tether, but it is not shown in any figure.

2) In the subsection “The nucleotide binding sites”, the entire fourth paragraph describes specific interactions between ORC proteins and ATP. However, none of these interactions are shown in figures, supplemental or otherwise. What is this analysis indicating for the reader? Why is this paragraph so important for the paper?

3) In the last paragraph of the subsection “The nucleotide binding sites”. Unclear whether the authors are referring to the R-finger or tether. To expand on this point, in the discussion of the "trigger" it would be helpful if it were explained what this means, and why the arginine presented across the subunit interface is not referred to as the trigger.

4) In the subsection “ORC ATPase levels are altered in Meier-Gorlin Syndrome mutants”, the authors argue that the ORC4-Y174C variant is hyperactive in the context of the motor module. However, this hyperactivity is invoked as a result of a single data point at high ATP concentration. How robust is this observation? This is confusing because this variant seems to have the opposite effect in the context of the ORC1-5 complex, and for the motor module the variant causes lower ATP hydrolysis at low [ATP].

5) The authors use the clamp loader structure to model DNA into the complex structure. How was this superposition performed? This point is important because superpositions onto different subunits (ORC or clamp loader) would result in slightly different orientations of DNA. Why did the authors choose this particular superposition?

6) In the last paragraph of the subsection “The DNA-binding site of HsORC”, the authors' structural analysis predicts a footprint of ~20bp. However, the yeast ORC complex has a footprint measured to be 88bp. To explain this discrepancy the authors propose a model in which the ORC complex greatly elongates upon ATP hydrolysis to extend its footprint from 20bp to 88bp. While an extension after ATP hydrolysis is feasible (and likely given the precedent of other AAA+/DNA complexes), it seems highly unlikely that the footprint of one complex is extended by over 4-fold. This would require that each subunit would protect ~15 bp, instead of the more typical 2-4bp seen in other AAA+ machines. This explanation seems so far-fetched as to be unreasonable. Please explain.

---

## [Author Response]

*Essential issues to fix regarding the electron microscopy:*

*The reviewers are concerned that there is something seriously wrong with the EM analysis, as the reconstructions represented in the figures do not square with the statistics. The only figure shown for the cryoEM structure is Figure 3 which is patently not at 9.9 Å resolution – it appears to be more like 20 Å, at best. At 10 Å resolution hints of α-helices should be visible in the density. It is also puzzling that the 2D classes in the Figure 3 supplement show detail that would be consistent with a 10 Å structure. How do these 2D classes relate to the final model? How many particles are averaged here and are these the from the final 10,000? There needs to be considerably more information to assess this and the model they present especially with respect to the Berger model of the inactive form of the complex.*

We regret not providing sufficient supplemental materials in our original submission. We also recognize that the EM map was rendered at a very low threshold in the original Figure 3, which has probably led to many if not most of the concerns. We have now included the requested information in the revised manuscript.

*From the discussion in Methods it seems they started with 530,000 particles then discarded 80% of those. The remaining 100,000 particles were divided into 6 classes and they then chose a class with just 10,000 particles, which can't even have been the largest class. This means the cryoEM structure actually represents less than 2% of the primary particles. Consequently, there is little confidence as to what this subset might represent. If it were representative of a selected subset that were, for example, a different conformation, it would still require considerable care and biochemical backup to validate this selection. A concern is that the majority of the particles may be "distressed" particles that are missing subunits, etc. The image shown in the supplementary would support this view as there seems to be many small particles that are probably partial complexes. Under such conditions, one needs to take great care (and have significant justification) in selecting out such a small subset for detailed analysis without describing what the other particles might represent.*

We agree that the raw micrographs, 2D and 3D class images all point to the fact that the sample may be “distressed” – there were many broken sub complexes, likely ORC145 and ORC23, and there was some ice contamination. Ideally, we would have liked to continue to optimize our samples and collect new datasets. However, despite these shortcomings, we are still confident in the 3D map because of the quality of the 2D class averages and the fact that several homolog ORC structures – *Drosophila* and yeast – have been described in the literature, which helped with selecting the right 2D and 3D classes. As the reviewer rightly noted that our goal here is not to obtain a high-resolution structure, rather, we are only trying to obtain a basic shape for docking of the two large subcomplexes of ORC1/4/5 and ORC2/3.

*The reviewers recognize that it can be justifiable under some circumstances to use only 10k out of 100k particles, but the micrograph in Figure 3—figure supplement 1 looks quite messy. If this is a 'representative' one, then the auto-picking could have picked up a lot of bad (clung together?) particles and only a small subset would be suitable for 3D reconstruction. Is this what happened? Have the authors considered the implications of this carefully?*

We agree with the reviewer. We understand that selecting a small percentage of particles can be risky for a new structure without any prior structural knowledge. But in our case, we were guided by the published structures of the highly conserved yeast and *Drosophila* ORC complexes.

*The 2D classes do look more detailed, but apart from two very similar views of the complex, only some classes which show what looks like partial-complexes are shown. Perhaps the angular distribution is very uneven and no sound 3D map could actually be calculated? An angular distribution plot would answer that question.*

ORC is largely a planar structure. It looks large in side views, but rather small in top and edge-on views. The smaller views that we have included are not from broken complexes but in fact belong to the intact complex. We have provided the angular distribution plot in our revision. The distribution is not even, and there are clearly some preferred orientations, as is the case for many published cryo-EM structures. But the angular distribution was sufficient to produce a medium resolution map. We agree that the resolution would have been better if the angular distribution was more even.

*The Fourier Shell Correlation (FSC) curve (Figure 3—figure supplement 1) is problematic. FSC curves of good cryoEM maps hover around 0.8 to 1 for most of the resolution range and then dip steeply near the resolution limit. This raises concerns regarding overfitting noise. The Methods do not mention anything about map validation. A stringent test would be to randomize all phases above 20 or 30 Å and then see if they come back during refinement. This will almost certainly show that the actual resolution is somewhere between 20 and 30 Å.*

We think the fact that the FSC value starts to drop so early (at ~ 1/30 Å) is due to the uneven angular distribution. Although the curve does not dip below the 0.143 threshold until 1/10 Å, there is very little information in the 3D map in the range of 1/10 – 1/14 Å range. In the revised figure, we have added a 0.5 threshold to indicate that the real information content may only reach to around 1/14 Å. This is also noted in the revised Methods and we changed the resolution reported in the main text and Table. Furthermore, we used a lower-pass filtered map (14 Å) for rigid body docking of our crystal structures.

*To summarize our position on the EM analysis:*

*There is a significant concern that the complex was either partially degraded (most likely) or that there were significant impurities in the preparation that gave rise to the mix of particle sizes. Given this, the cryoEM reconstruction needs to be carefully validated, and presented at an appropriate resolution. In reporting on their data after reanalysis, the authors should pay attention to the following points:*

*1) The authors should provide a display of all 2D classes, not just the best ones. The same is true for the 3D classes – all six should be shown.*

In revision, all reference-free class averages that were used to select/include raw particles are displayed. Six 3D classes are also included in Figure 3—figure supplement 5.

*2) The authors mention an initial study using negative stain. These data should be shown, including the 3D reconstruction they obtained from these data. Furthermore, they should explain how they obtained this first 3D reconstruction. Finally, was this 3D reconstruction used to initiate processing of the cryo-EM data using Relion? If not, which initial reference did they use?*

A representative negative stain raw image, 2D classes, and 3D map are included as new supplemental figures – Figure 3—figure supplements 2 and 3. We have explained in the revised methods section that we used a low-pass filtered (55 Å) yeast EM map as the starting model to arrive at the negative stain human ORC1-5 map, as hORC1-5 in 2D averages resemble that of the yeast ORC.

*3) The authors should show examples of the initial 3000 particles picked manually and the 2D class averages obtained from them that were used as templates for subsequent automatic picking. It is quite possible that this initial picking step biased the rest of the picking to favor particular views and exclude other views of the complex.*

We have included all 2D classes that were used to include raw particles. The smaller size in some classes are not broken particles, they are tilted or ‘near-top’ views.

*4) The authors should also explain what they think is "wrong" with the majority of particles that they call "bad".*

As shown in Figure 3—figure supplements 4 and 5, the “bad” particles were excluded at two stages – they either belonged to 2D class averages that are featureless blobs, or to the 3D classes that appear to be broken structures or don’t look like the ORC structure. Such selection was made possible by the existing structural knowledge of the well-conserved yeast and *Drosophila* ORC.

*5) The authors should show a random sampling of particles from the 10,000 particles that were used for the final 3D reconstruction, together with matching projections to show that the alignments were plausible and the 3D reconstruction is consistent with these particles.*

The truncated ORC particle is comparatively small at about ~ 200 kDa. The enlarged raw particles have little visually recognizable features for comparison with averaged images or re-projections, and therefore, are not shown. Nevertheless, two raw cryo-EM micrographs are included in the supplemental figures.

*6) In the present manuscript, the presentation of the cryo-EM structure of HsORC1-5 in Figure 3 indicates poor fitting of the ORC2 model to the density map, and there is no EM density map superimposed onto the structure in the orthogonal view shown in Figure 3. The authors need to describe how good or bad the fit is, once the EM reconstruction is validated.*

An orthogonal view is presented as a supplementary figure (Figure 3—figure supplement 8) as well as a movie with a full rotation. We did not feel the resolution of the EM map warrants flexible fitting. However, we have a high degree of confidence in the crystal structures, especially that of ORC1/4/5. ORC1/4/5 fits well as a single rigid body into the density.

As mentioned in the text, we used ORC2/3 as a second rigid body, along with the homology model of HsORC5-WHD, using a similar interface to that in *Drosophila*, and finally docked the ORC2-WHD.

Essential revisions concerning other aspects of the paper:

*The reviewers are concerned that the paper does not place this work properly in the context of the previous paper from Berger on the structure of the Drosophila ORC complex. The Introduction should begin by providing the context, outlining what is known previously at a structural level.*

We now refer to the structure of *Drosophila* ORC, as well as the low resolution EM structures early in the Introduction of the manuscript.

*The authors do not discuss how their structure relates to MCM loading mechanism in sufficient detail. The architectural similarities to the sliding clamp loader are noted and would naturally lead to some hypotheses for how ORC loads MCM. However, the authors don't present possible mechanisms that their structure naturally points to.*

Although it is not known exactly how ORC and CDC6 cooperate to load a circular MCM2-7 hexamer onto double stranded DNA, we now discuss at the end of the Discussion section some ideas about this process based on the current structure and known biochemistry using the yeast system.

*More specifically, references to the Berger structure are scant and it seems the authors describe aspects of the structure that are the same or at least very similar, which should be acknowledged. The first concession that there is any similarity is in the subsection “HsORC1 exists in an active conformation”, when they only mention the Berger work to say the architecture of the complex is similar but the conformation is different, although they provide no figure for the comparison to be assessed.*

A comparison between the structures certainly exists already in the original manuscript. A specific paragraph regarding the differences between ORC1 in human and fly is included. In addition, a comparison between ORC1 from human and fly are illustrated in Figure 2 and it’s positioning in the complex in Figure 3 in the main text. Two of the four figures in the main text include a comparison with the fly structure. In addition, we included an animation illustrating a morphing between the two structures as supplementary information, for the reader to assess the comparison more closely.

However, in the revised manuscript, we refer to the fly ORC in the introduction. We added the use of portions of DmORC as a search model for structure determination in the opening of the section describing the structure of the HsORC motor module in the main text in addition to the Methods section where it was already described in more detail. The use of the DmORC structure for HsORC2-ORC3 is already described in the main text of the original submission.

We now also add an additional figure of a side-by-side comparison of the two ORC complexes (Figure 4).

*In fact, the Berger coordinates were used for molecular replacement, although the methods do not say whether they were used as a single unit in each case (Orc1, 4, 5 and Orc2, 3) or the subunits were found separately. We presume the latter, since the paper suggests the conformations are different although the only comparison of the structures is of one subunit in the panel in Figure 2. The cursory nature of the references to the Berger paper is inappropriate, when it is evident that the similarity is in fact significant.*

We are surprised by this comment since in the original manuscript the exact residue numbers for the MR search model was provided (see second paragraph of Protein Crystallization and Data Collection section for ORC1/4/5 and in the fifth paragraph of the same section for ORC2/3). Nevertheless, as stated above, we added a clear statement regarding the use of the fly structure in the main text in the opening sentence of the description of the structure of the motor module, and a statement regarding the use of this structure in the section describing ORC2/3 was already present. A statement regarding the similarity between ORC4 and ORC5 of the two complexes was also already present in the original manuscript. However, now we have also added an RMSD between the ORC4-ORC5 subunits in fly vs. human, to underscore the similarity. As also stated above, we added a figure with a side-by-side comparison of the more complete complexes.

*The important difference is that the present work has crystal structures (of sub-complexes) that contain bound nucleotides.*

It would have been nice and straightforward if the presence of nucleotides had been sufficient to explain the difference in the states between the two structures. However, the Berger paper refers to another structure, which includes ATPγS that has a similar conformation to the structure devoid of nucleotides, thus a similar inactive conformation. They even show electron density maps for the nucleotides, albeit with somewhat weak density.

*However, it is well established in these AAA systems that it is essential to have details of the strain induced around rings or spirals when nucleotides are bound, as this creates conformational changes that drive the biochemical processes. Clamp loading is one pertinent example since they draw several comparisons with that system. The explanation for now nucleotide binding induces the conformational changes observed is not well explained, and should merit additional discussion.*

We agree that AAA+ proteins change conformation in response to nucleotide hydrolysis, but in the case of ORC-CDC6, exactly how this happens in still not resolved, even in the better characterized yeast system. The high-resolution structure provided here will be a valuable guide to designing further tests of the ATPase activity. In this regard, the Meier-Gorlin mutation in ORC1 described here will be a valuable reagent.

[Editors' note: further revisions were requested prior to acceptance, as described below.]

*[…] Conclusion: We believe that the cryo-EM work, as presented, has flaws and is of low quality. However, given the negative-stain structure and assuming that there is some truth in the cryo-EM structure at ~20 Å resolution, the data, processed appropriately to lower resolution, might be sufficient for what the authors want to show: activation of ORC results in an enlarged cavity for DNA and that there is a gap in the AAA ring that can be filled by ORC6.*

*Alternatively, given that the negative-stain reconstruction is actually quite good compared to the cryo-EM reconstruction, the authors may wish to include that either in the main analysis or it in the deposition of structures in the database.*

*The revised manuscript should include in the main text a discussion of the EM issues (e.g., those highlighted above) and the approaches taken to include a reliable interpretation in the final analysis. The discussion should be such that an interested reader is left with a good sense as to the extent to which the reconstruction should be trusted. It is essential that the reviewers are reassured on these points.*

We thank the reviewer for the suggestions. In the revised text, we note: “The purified HsORC1-5 described here is a fairly unstable complex, consistent with the dynamic assembly and disassembly of HsORC in vivo. […] Thus, we reasoned that using this subset of particles in order to obtain a low-resolution structure into which we could fit the crystal structures of the subcomplexes (ORC1/4/5 and ORC2/3) was warranted.”

We agree with the reviewer that the negative stain map looks quite good, because the particle appeared stable and had good contrast in negative stain. But the 3D map is slightly distorted, probably due to the flattening effect by the staining procedure or binding on the carbon film, such that the map does not fit well with the crystal structure in the ORC2/ORC3 region. However, the cryo-EM, derived using the staining map as a starting model, fits much better to the crystal structures. We therefore present both negative stain and cryo-EM maps in the manuscript.

To address the reviewer’s concern on the very small percentage of the particles included in the final dataset, we have 4x binned the particles to limit the information used in particle alignment. Refinement of the binned dataset resulted in a 3D map with an estimated resolution of 18 Å. We further explained in the text that the FSC curve dips well before 18Å, so the map may only contain reliable information up to ~ 20 Å. And we listed the resolution in Table 2 as 20 Å rather than 18 Å.

We believe that this resolution is satisfactory for what we wanted to achieve here – obtain a suitable map into which we could fit the independently obtained crystal structures. As can be seen, the crystal structures fit extremely well into the EM map. We believe that the EM data is satisfactory for this very unstable complex.

*Manuscript text and figures:*

*We feel that the manuscript, as written and illustrated, falls short of presenting the work in a way that properly reflects the importance of the results. The current manuscript risks losing the attention of readers who are not well versed in the topic.*

*Please consider the following points seriously in preparing a revised version:*

*1) The Introduction is striking for its brevity, and it fails to set the stage for the general reader to appreciate the results. It consists of two paragraphs. The first is a very long run-on paragraph in which many concepts are elided. The second paragraph describes the crystal structures, but provides no introduction to the EM analysis. The Introduction should be rewritten with an eye to clarity and also to maximizing appreciation of the results.*

We have revised the Introduction extensively and highlighted the importance of this work in light of what is known in the field. We discuss the previous work in more detail and provide clearer explanations regarding the activity/inactivity issue. We have also highlighted in the new Introduction the surprising structural similarity between ORC-CDC6 and the RFC-like DNA polymerase clamp loaders, and now refer to the EM analysis.

*2) The figures could be improved substantially. The views of the structures could be augmented by showing an alternative orientation in which the nucleotide coordination is emphasized rather than the overall assembly architecture. The labeling of the figures is sparse, causing the reader to strain to understand all the components. We have no set limits – take the space that you feel is appropriate to illustrate your results to best effect.*

Figures with new views were added in the main text and as supplementary figures. Labels were also added:

Figure 1: new view and labels added.

Figure 1 was also split so that the old Figure 1 is now Figure 2.

Figure 3 (old Figure 2): expanded view for Figure 3; new labels on Figure 3

New Figure 4 (old Figure 3): and added new labels.

New Figure 3—figure supplement 1.

New Figure 4—figure supplements 5, 6, 7 and 8 (and added new labels).

*3) In comparing the structure obtained here to other structures, it is very difficult to judge changes in inter-subunit orientation. For example, in Figure 1, there is no superposition of ORC with RFC to show that the inter-subunit orientations are preserved. Likewise, one cannot readily tell that the inter-subunit orientations are preserved within ORC itself.*

It would be very difficult to show a superposition with the clamp loaders because of their multi-subunit nature. We think that the side-by-side figure depicts the similarities quite nicely. We think that the similarities are fairly obvious. We also don’t mean to imply that they superimpose perfectly. This is not the case even within the RFC family. For example, the relative orientation between the subunits within the RFC family range between 56° and 74° in rotation and 2.8 to 8.0 Å in rise, excluding a couple of clamp loaders that are in a non-active conformation, which deviate even more significantly.

Therefore, the differences between these rotations within ORC are smaller than the range between clamp loaders. We added the precise numbers for ORC, the ranges for RFC and that ORC is on the shallow side. We also noted in the figure legend that the alignment was based on the subunit depicted in cyan (ORC4 in our case).

*4) The revised manuscript mentions the crystal structure of DmORC in the Introduction, but categorically labels it as "inactive" without giving any explanation as to how "inactive" is defined or describing what the structure includes. (We appreciate what you mean, but we fear that the general reader would not.) Following a too brief discussion of the DmORC structure, the authors state that "we report the structure of human ORC in a functionally active state". But their crystal structure is of a three-subunit sub-complex and not the six-subunit human ORC. The "activity" refers to only ATP hydrolysis by the sub-complex and not initiation of DNA replication. Some explanatory text could tie this all together and present the new work in the best light.*

We expanded the Introduction and discussion regarding ATP hydrolysis by ORC, the importance of the different ATP sites and clarified what we mean by “active” and “inactive”. We also expanded the discussion on the structure of DmORC, so that our contribution is better appreciated in light of the known structure. The structure is not just the active, three-subunit motor domain, but we describe the structure of the 5-subunit complex and measure ATPase activity of the entire 5-subunit complex.

*Additional important points:*

*1) In the second paragraph of the subsection “The nucleotide binding sites”, the authors discuss ORC5-R104 as the tether, but it is not shown in any figure.*

We added a supplementary figure showing this interaction.

*2) In the subsection “The nucleotide binding sites”, the entire fourth paragraph describes specific interactions between ORC proteins and ATP. However, none of these interactions are shown in figures, supplemental or otherwise. What is this analysis indicating for the reader? Why is this paragraph so important for the paper?*

Since we are describing the ATP binding sites, we included the contact with the base for completeness. We have added supplementary figures to this revision to illustrate these contacts.

*3) In the last paragraph of the subsection “The nucleotide binding sites”. Unclear whether the authors are referring to the R-finger or tether. To expand on this point, in the discussion of the "trigger" it would be helpful if it were explained what this means, and why the arginine presented across the subunit interface is not referred to as the trigger.*

There has been a bit of confusion over the years regarding how the three basic residues opposite the Walker A and Walker B are named. There is usually an R-finger, but this is different in different ATPases – most notably the R-finger of the F1 ATPase refers to the first arginine, whereas in some of the helicases the R-finger is the middle one. However, it appears that their position is linked more appropriately to their role in ATP hydrolysis in these types of molecular machines than the names they were initially given. We therefore tried to sort this out a few years ago, so as to be able to be more explicit when referring to these residues across the different types of these multi-subunit ATPases. We named these basic residues according to their role. This was published a few years ago in a review we wrote. However, since this is not widely used yet, we did explain what we meant by “trigger” etc. in the subsection “The nucleotide binding sites”, nevertheless, more was added for additional clarification. We also added labels in Figure 2. In the case of basic residues on ORC3 for the ORC5-ORC3 interface, we clarified what we are referring to in the text.

*4) In the subsection “ORC ATPase levels are altered in Meier-Gorlin Syndrome mutants”, the authors argue that the ORC4-Y174C variant is hyperactive in the context of the motor module. However, this hyperactivity is invoked as a result of a single data point at high ATP concentration. How robust is this observation? This is confusing because this variant seems to have the opposite effect in the context of the ORC1-5 complex, and for the motor module the variant causes lower ATP hydrolysis at low [ATP].*

The observation is indeed robust and was therefore included. As seen in the figure, the error bar is very small and this result is reproducible with different protein preparations. We noted already in the text that the effect is different in the context of ORC1-5. It appears that ORC2/3 have the effect of keeping the complex in check.

*5) The authors use the clamp loader structure to model DNA into the complex structure. How was this superposition performed? This point is important because superpositions onto different subunits (ORC or clamp loader) would result in slightly different orientations of DNA. Why did the authors choose this particular superposition?*

We elaborated on the exact way this was performed in the main text, as well as the reasoning behind our choice of T4-gp44.

*6) In the last paragraph of the subsection “The DNA-binding site of HsORC”, the authors' structural analysis predicts a footprint of ~20bp. However, the yeast ORC complex has a footprint measured to be 88bp. To explain this discrepancy the authors propose a model in which the ORC complex greatly elongates upon ATP hydrolysis to extend its footprint from 20bp to 88bp. While an extension after ATP hydrolysis is feasible (and likely given the precedent of other AAA+/DNA complexes), it seems highly unlikely that the footprint of one complex is extended by over 4-fold. This would require that each subunit would protect ~15 bp, instead of the more typical 2-4bp seen in other AAA+ machines. This explanation seems so far-fetched as to be unreasonable. Please explain.*

We agree with the reviewers and changed the discussion regarding the possible role for ATP hydrolysis by ORC. This is now described in a new subsection.